# A Comprehensive “Real-World Constraints”-Aware Requirements Engineering Related Assessment and a Critical State-of-the-Art Review of the Monitoring of Humans in Bed

**DOI:** 10.3390/s22166279

**Published:** 2022-08-21

**Authors:** Kyandoghere Kyamakya, Vahid Tavakkoli, Simon McClatchie, Maximilian Arbeiter, Bart G. Scholte van Mast

**Affiliations:** 1Institute of Smart Systems Technologies, Universitaet Klagenfurt, 9020 Klagenfurt, Austria; 2P.SYS System Creation KG, 9500 Villach, Austria

**Keywords:** activity monitoring of “humans in bed”, abnormal behavior detection and forecasting, comprehensive anomaly concept definition, explainability and interpretability of detected anomalies, uncertainty modeling, early warning capability, behavior evolution identification and system adaptivity, verification and validation, anomaly detection and prediction, scenario analysis, selected use-cases, user perspectives of practical interest, real-world context-aware and practically realistic specification book, comprehensive critical state-of-the-art review, system architecture and system engineering related recommendations

## Abstract

Currently, abnormality detection and/or prediction is a very hot topic. In this paper, we addressed it in the frame of activity monitoring of a human in bed. This paper presents a comprehensive formulation of a requirements engineering dossier for a monitoring system of a “human in bed” for abnormal behavior detection and forecasting. Hereby, practical and real-world constraints and concerns were identified and taken into consideration in the requirements dossier. A comprehensive and holistic discussion of the anomaly concept was extensively conducted and contributed to laying the ground for a realistic specifications book of the anomaly detection system. Some systems engineering relevant issues were also briefly addressed, e.g., verification and validation. A structured critical review of the relevant literature led to identifying four major approaches of interest. These four approaches were evaluated from the perspective of the requirements dossier. It was thereby clearly demonstrated that the approach integrating graph networks and advanced deep-learning schemes (Graph-DL) is the one capable of fully fulfilling the challenging issues expressed in the real-world conditions aware specification book. Nevertheless, to meet immediate market needs, systems based on advanced statistical methods, after a series of adaptations, already ensure and satisfy the important requirements related to, e.g., low cost, solid data security and a fully embedded and self-sufficient implementation. To conclude, some recommendations regarding system architecture and overall systems engineering were formulated.

## 1. Introduction

Abnormality detection and/or prediction is currently a topic of very high interest. In this paper, we addressed it in the frame of the activity monitoring of a human in bed. Indeed, humans typically spend a significant portion of their daily lives in bed. This time becomes even longer in cases where the human is unwell. This is particularly the case for sick or older people, who spend even more time in bed. Their physical activity or inactivity patterns provide useful signatures that reflect the “state” of the person under observation. In the frame of activity monitoring endeavors, behavioral situations that are abnormal (these situations are more/extremely rare within the observation time window) are the ones that are of the highest interest when compared to behavioral situations that are rather normal (these ones occupy most of the observation time window).

Monitoring humans in bed, e.g., in the context of “sleep monitoring”, is very important for a series of human-specific health conditions. For elderly people, for example, inadequate and irregular sleep (which can be inferred from physical movements on the bed while sleeping) is often related to serious diseases such as depression and diabetes. Indeed, in several cases, it is necessary to monitor the body positions and movements made while sleeping (or just while lying in bed) because of their relationships to either particular diseases (i.e., sleep apnea and restless legs syndrome) or particular anomalous behaviors of relevance w.r.t. the specific observation context of the human in bed. Analyzing movements (or, more generally, physical activities) during sleep can also help in determining both sleep (or laying down) quality and irregular sleeping (or laying down) patterns. 

Lying in bed, especially for longer times or more than only some minutes, is generally motivated by the need to rest due to several health status-related contexts. For example, one is sick (different types of sickness, different levels of sickness), one is very tired, one is very old and weak, one is weak or tired, a lady one day or several hours before giving birth, a person in rehabilitation after either a chirurgical operation or a stroke, etc. 

Moreover, it is well known that sleep plays an important role in the quality of life and contributes significantly to staying healthy, active and energetic. In special residences such as in the so-called nursing and retirement homes, periodic observation rounds (of the medical or nursing personnel) during the night are a major disruption for the residents and can cause distress and sleep deprivation. Thus, some intelligent technical system capable of reliably performing the monitoring endeavor of those named residents is most welcome. 

Overall, it can be stated that sleep monitoring systems, to name a few illustrative use-cases, enable the recognition of sleeping disorders as early as possible for diagnosis and prompt treatment of diseases. Such smart monitoring systems can indeed provide healthcare providers with quantitative data about irregularities (in related positions and movements in bed) in sleeping periods (or more generally in laying periods) and durations. They can also provide detailed sleeping/laying profiles that depict periods of restlessness and interruptions, such as bed exits and bed entries due to either visiting the bathroom or performing other activities in the home.

Numerous sensor technologies exist that can be involved in the acquisition of data from a bed w.r.t. to movement and/or positions of the human lying in the bed. There are several monitoring devices available on the market that are used for sleep tracking or for more safety during the night. These sensor systems can be divided into wearables, such as smart watches and fitness trackers, sleep monitoring belts and devices that are placed on or under the mattress [1,2,3,4,5,6]; devices that are attached to the pillow [2]; smart bed sheets, mattresses and pillows [2,7,8]; sleep monitoring devices that are placed beside the bed [9]; and camera-based systems in the room [10]. Furthermore, there are monitoring systems available and capable, for example, of measuring parameters of sleeping babies [11] and people with epilepsy [12]. Most of those devices are mainly configured as lifestyle gadgets for rather private applications. However, some manufacturers have already been offering special solutions for care institutions [3,4,8,10]. Thereby, most existing sensor systems use piezoelectric sensors, accelerometers and/or radio-frequency identifiers to detect a series of relevant parameters. Indeed, the use of radio-frequency identifiers [13,14], air-pressure sensors [15,16], smart textiles [17,18], photoplethysmography [19] and thermopiles [20] has already been extensively researched, as described in several scientific papers. Some of those papers also discussed the use of piezoelectric sensors [21,22] or load cells [23] below the legs of a bed to monitor the parameters of a person lying in bed.

Sensors placed under each of the four bed legs are a representative example of the various sensor systems described in the previous paragraph. These types of sensors are particularly capable of measuring variables related to all forms of physical activity in the bed, e.g., through weight variations or motion detection. A monitoring system involving the data generated by such sensors can capture different activity-related physical variables through the use of more than one sensor type under each bed leg [24]. The following are some examples of usual physical activities in the bed for which related variables can be detected by the various sensors (non-exhaustive list): sitting on the bed with feet on the ground, standing up from the bed with feet on the ground, sleeping on the bed, turning oneself in the bed, sitting in the bed (with feet in the bed), etc. 

The comprehensive and holistic discussion of the abnormality concept in the overall observation context of the “monitoring of a human in bed” is motivated by a series of insights originating from a critical analysis of the relevant literature. First, overall, one realizes that there is a big and serious confusion regarding a comprehensive and universally solid, valid and accepted definition of the concept “anomaly”. What some authors call or understand under “anomaly” comes rather too close to other concepts of relevance such as “activity”, “event”, “abnormal behavior” and “behavior change”. Moreover, the “anomaly” definition becomes more complex in view of the fact that the system under observation (i.e., a human in bed) is in various system-related aspects very different from a pure technical machine as usually addressed in most of the relevant literature. It is also of special relevance that the context “human in bed” is itself a relatively diverse and broad universe where various fully different and interesting monitoring use-cases may be encountered. For example, alone, the specific human type/sample under observation may make a huge difference in the case settings. The following few examples may underscore this sensitive nuance: (a) the human under observation is a one-year-old baby or a small child between two and five years; (b) the human under observation is a healthy athletic fit young man; (c) the human under observation is a very old (90 years old) and sick person in a coma; (d) the human under observation is a pregnant woman awaiting to give birth within a couple of days, etc. One can assume from common knowledge that the physical activity patterns and their respective dynamic evolutions over time (short-term, average-term and long-term) for these human types are fully different from each other. Thus, the unsupervised-learning-related “anomaly” definition shall be robust w.r.t. those evident differences amongst different human samples under observation. Moreover, one can also ask whether the “anomaly” definition shall be “black” (meaning it assumes unsupervised learning or identification), “white” (meaning it assumes an expert-supported (or via a reference labeled dataset) supervised learning or identification), or “gray” (meaning a merging of “black” and “white” identification).

From the broad anomaly literature in the contexts of machines’ or technical systems’ observations, one learns that there are various types and forms of anomalies. Thus, just stating or detecting the presence/occurrence of an anomaly is not sufficient at all. One shall also specify the related anomaly type and/or form. An interesting question for this review study is whether all of those anomaly types and forms observed in the context of machines’ observations are also relevant and observable for our core context, “monitoring of a human in bed”.

Even after the “anomaly” concept has been well defined and assessed regarding type and form, another very sensitive and practically relevant dimension is its location of it in time. In any observation/monitoring endeavor, the time dimension fixes three regions, whereby each of them can also be divided into respective sub-regions: (a) past, (b) present/now/current and (c) future. If we consider the future, sub-regions can be near-future, middle-future and far-future. The specific measure of each of these sub-regions (for an illustration of the future) may be very dependent on the setting of the specific observation case. The “near future” may be some seconds, some minutes, some hours, or even some days, depending on the given human observation use-case. This suggests that the use-case engineer shall specify the various use-case related realistic/appropriate boundaries of the different time dimension-related sub-regions. Therefore, a given well-defined “anomaly” shall be placed in a specific sub-region of the time dimension. A very interesting question of relevance for science is that the detection of anomaly may happen/occur or be confirmed at various relative times. The relativity referred to here is the one between the time of occurrence (or time sub-region) of an anomaly and the time it is effectively detected/assessed/predicted.

Indeed, a currently running anomaly detection algorithm can produce a result related to three different cases in their meaning, interpretation and relevance for the specific monitoring use-case:**Time Case 1—Posterior anomaly detection**: the anomaly happened in the past (in one of the different sub-regions of the past), but it is detected now. Here, the detection of posterior is the anomaly occurrence;**Time Case 2—Online or simultaneous detection**: the anomaly happens in the current time, now (in one of the different sub-regions of now), and it is detected now. Here, the detection and anomaly occurrence are both in the current time, now;**Time Case 3—Forecasting or anterior anomaly detection**: the anomaly will happen in the future (in one of the different sub-regions of the future), but it is detected now. Here, the detection of anterior is the anomaly occurrence.

As the underlying (prior) knowledge supporting each of these three cases (Case 1, Case 2 and Case 3) is different for each of them, one shall involve fully different detection schemes and approaches. The respective results also have different meanings and relevance for the specific human monitoring use-cases. Moreover, at least theoretically, the same observation data that are judged “anomaly” may later (i.e., posterior) be differently viewed as normal, and vice-versa. This shows a specific aspect of the anomaly-related time dynamics of observed situations; that is, a modification of the labeling of the same “observed situation” as time passes. Indeed, “behavior change” is one of the complex forms of the “anomaly” concept, which may explain such complex labeling dynamics.

The previous differentiation of cases is sure of high practical relevance. Let us briefly consider the data processing-related hierarchy and perspective. Essentially, the available sensors produce “raw data”. Traditional machine learning suggests the following data-processing-related hierarchy w.r.t. the gradual processing levels. From “raw data” (Level 1), one extracts or generates “features” (Level 2). From features, one generates “low-level events” or “activities” (Level 3). From low-level events, one generates/extracts “high-level events” or situations (Level 4). In principle, an “anomaly” may be located at each of the listed data processing hierarchy levels: Level 1 to Level 4. Several anomalies at the lowest levels (Level 1) of this data-processing hierarchy may be due to sensor-related problems such as “missing data”, electric disturbances in the sensors, sensor system’s electronic faults, outliers, etc. Most of these anomalies at the lowest level (since they are mainly due to sensor issues) are of no significance for most human (in bed) monitoring use-cases and shall be filtered out and addressed by some appropriate pre-processing modules of the raw sensor data.

Thus far, we have identified various complexity dimensions of the anomaly concept in general and for our core context (monitoring of a human in bed) in particular: anomaly types and forms, observation subject dependency, use-case dependency, occurrence time-related location, relative timing between occurrence and detection, data processing level, etc. If all that complexity is not sufficient, several modern use-cases come with a particularly hard although practically very useful characteristic and requirement w.r.t the anomaly concept: “Anomaly Explainability”. In many use-cases, the anomaly of interest shall be the ones located at the highest data processing level, namely at Level 4. The basic bricks of the explainability of anomalies are founded on explicitly modeling the complex dependencies between different “variables”, which significantly contributes to the ability to detect anomalous occurrences at a given processing level. Those named “variables” may be entities of the same and/or lower processing levels (normal or abnormal) or entities of the same or different time regions or sub-regions. This is valid for all three time cases described above (posterior, simultaneous, anterior). This explainability dimension is part of the so-called “Explainable Artificial Intelligence (AI) ” and is very crucial for the practical user acceptance of monitoring systems of humans in bed. It enables one to explain (it provides coherent reasoning) why and how the anomaly happens. It is a set of processes and methods that allows human users to comprehend and trust the anomaly detection results created. Indeed, “Explainable AI”, according to the literature, is used to describe any given “intelligent” model, its expected impact and potential biases. It helps to characterize model accuracy, fairness, transparency and outcomes in “intelligent system”-powered decision making. Most human (in bed) observation use-cases clearly need this capability. “Explainable Anomaly” is therefore very crucial for the target user organizations (examples: hospitals, care facilities for older people, etc.) in building trust and confidence when putting “monitoring systems for human bed” into production.

Motivated by the general background just provided, this review and communication paper intended to provide the following essential contributions; each of these contributions was addressed in one or more of the next sections of this paper:Briefly describe the basic settings and the general architecture of the monitoring context of a human in bed;Identify the currently poor definition of the “anomaly” concept and suggest a comprehensive and coherent definition that is consistent with systems engineering-related practical requirements. A comprehensive definition of the anomaly concepts shall include all important dimensions of its complexity. A clear differentiation to other closely related and/or coexistent concepts is provided;Furthermore, provide a comprehensive explanation of a series of relevant characteristics of a robust and real-world mature anomaly detection endeavor. For example, the following: subject-dependency; use-case dependency; activity/event attributes/features; anomaly score; uncertainty quantification; unsupervised learning versus semi-supervised learning versus supervised learning; the complex and time dynamic and/or structural relationship between events (at/of different levels of the data-processing hierarchy), events sequences and anomalies; the relative timing between detection and occurrence; the observation perspectives of the anomaly phenomenon, which are, amongst others the following: (a) “condition/entity/event-type related anomaly”, (b) “time-window related anomaly”, (c) “multiple time-windows related anomaly”, (d) “conditions/entity/event-sequence related anomaly”, etc.; adaptivity based on data aging, data dissimilarity forgetting factors and novelty detection;Formulate a comprehensive requirements engineering dossier for a robust and practically useful anomaly detection system. A subsequent representative specification book shall be presented. The specification shall distinguish three classes of requirements: the MUST-HAVE requirements, the NICE-TO-HAVE requirements and the NEVER HAVE/DO NOT requirements;Comprehensively discuss the verification and validation of the challenge of a robust and real-world mature anomaly detection system in the context of the monitoring of humans in bed. Hereby, amongst other concerns, the appropriate setting and availability of reference datasets are discussed;Then, a comprehensive critical review, identifying respective limitations, of how far the major schemes/approaches for anomaly detection as per the current state-of-the-art are capable or not to fully satisfying the major requirements described for the robust and real-world mature anomaly detection system;Finally, the suggestion and discussion of a tentative general and strategic architecture of a truly robust anomaly detection scheme, which has the potential of fully satisfying all of the MUST-HAVE requirements of the formulated comprehensive specification book. The bricks of this architecture, which have capabilities going beyond those of competing schemes from the current state-of-the-art (see also the reference system described in Section 2), are briefly discussed.

## 2. General Context Description of the Monitoring of Humans in Bed, One Illustrative Example from the Practice

A typical state-of-the-art representative monitoring system of humans in bed can be described as follows. A good example of such a system is the one developed by the company “P.SYS system creation KG” [25]. The aim of this monitoring system is to provide a user-friendly system that connects older people, or people who generally live independently, in the event of irregularities (in their daily behaviors) with help providers from their extended social environment in a timely and autonomous manner.

The core focus here is on the development and maturing of a bed monitoring system that detects irregularities in a person’s physical movement patterns while sleeping on a bed. For related sleeping patterns, the bed monitor collects data from sensors placed under the bed legs (see Figure 1). The signals form a superposition of body vibrations (movement, breathing, etc.), the weight distribution of the person lying in bed and external influences. The bed monitor can, respectively, be used in the care sector and subsequently also in private households in order to be able to detect exceptions that manifest themselves in deviations from a person’s normal sleeping behavior and to be able to react to them with an appropriate alarm.

Regarding the intelligent system implemented, it already realizes most of the features of the related state-of-the-art. Essentially, a model based on advanced statistical methods (such as hidden Markov models and others) was already developed, which independently learns the usual patterns in the data and reacts to exceptions. The developed algorithms work sequentially, with the observations being processed in real time. The data streams from the sensors are not stored in the model; only the model parameters are adjusted and stored using the observed data. Beyond detecting exceptions, one can also assess the quality (eventually in different quality levels) of the sleeping process and/or identify specific special events. This system can deal with subject-dependency, as it learns individual behavior and reacts to individual exceptions. It essentially learns from each individual user under observation on the bed (Figure 2).

## 3. A General Comprehensive Discussion of the “Abnormality Concept” in General and in Particular w.r.t. the Monitoring Context of Humans in Bed

Traditional home-based context-aware remote monitoring systems [26] use different techniques such as rule-based reasoning, probabilistic models, data mining, etc., which can detect abnormalities only when they occur in an ongoing situation. One shall note carefully that an abnormality happens within an ongoing (or future) situation. Thus, the situational dimension and concept and its related ontology must be comprehensively defined and described for the context of monitoring humans in bed. Several such systems have no option/capability to utilize long-term situational data for modeling each user state individually and predicting future anomalies ahead of time. It is worth underscoring this very practically relevant capability of monitoring systems, namely that of a long-term situational awareness and prediction of future anomalies. This capability refers to a special “complex early warning system” functionality. Monitoring systems that have this capability (notice that it is a very tough and very advanced requirement that represents the cutting-edge), provided that it is reliably and robustly validated (e.g., with a confidence level above 90% or better even up to 98% to be of immediate relevance for practical, real-world implementations [27,28]), are able to alert care-takers or any other relevant personnel or family member about the incoming changes before the situation becomes critical.

As already stated above, there is considerable and serious confusion regarding a comprehensive and universally solid, valid and accepted definition of the concept “anomaly”. The first semantic problem is that several authors suggest that “anomaly” is an independent entity, which is a wrong perception. Indeed, and more correctly, one shall rather state that an “anomaly” is always information, a judgment or a rating of one or more particular “behavioral entities” of the system under observation. A complex system under observation may have various types of entities operating/occurring at different system levels and/or from different system observation perspectives.

First of all, it is important to separate (assume they are pre-filtered from the analysis) the faults, signal perturbations and other imperfections related to the electronic measurement sensor system from the “behavioral entities” that are solely linked and part of the system under observation. Thus, we focused mostly on anomalies related to the intrinsic physical activity behavior of the human in bed under monitoring.

Assuming there is a consensus w.r.t. the understanding that “anomaly” is always information, judgment or a rating of one or more particular “behavioral entities” of the system under observation, one can now suggest the description of a comprehensive ontological and semantic framework of the anomaly concept. This framework considers a series of dimensions:Hierarchy levels (data-processing related) of the behavioral entities;Form-type related anomaly classification;Observation-context perspective related anomaly classification;Learning strategy and learning context-related perspective. Examples: “black” versus “white” versus “gray” versus “subject-dependent” versus “use-case dependent” modeling and identification of anomalies;Novelty-related anomaly classification (see “novelty” detection; see also continual learning);Subject-dependency-related anomaly detection.

### 3.1. Consider the Hierarchy Levels (Data-Processing Related) of the Behavioral Entities

We suggested the following hierarchy of data-processing levels, which is consistent with traditional machine learning pipelines; see Figure 3. We defined five core data processing levels, namely Level 0, Level 1, Level 2, Level 3 and Level 4.

**Consider Level 0**. The acquisition; structuring; and various combinations, abstractions and pre-processing of the raw sensor data are managed at Layer 0. Level 0 produces a multivariate dataset that enables Level 1 to better identify the first race of different entities (i.e., Level 1 entities), which are relevant for the overall system-related behavioral characterization. In Level 0, we did not define entities of relevance. Of particular interest at Level 0, one may nevertheless point out aspects such as grouping certain sensors, aggregating sensor data and extracting from the raw sensor data various “metadata” that are perceived to be useful.

**Consider Level 1**. This is the first level at which relevant behavioral entities were defined, identified and/or detected. At this level, the entities are called ACTIVITIES. The signature of a Level 1 entity is one of the possible/observed patterns provided by the Level 0 complex multivariate time series. If there is no supervision at all, the Level 1 entities are determined through fully unsupervised learning. Otherwise, semi-supervision or full supervision is always possible and welcome. Indeed, depending on the specific monitoring use-case, some of the Level 1 entities may be known or suggested from the given application context, thus enabling at least a semi-supervision. In the worst case, at least a fully unsupervised identification is always possible, the most naïve of them being a form of deep-clustering. As is discussed more in later sections, amongst others for laying the ground for the so-called ”explainability” (see the “explainable AI” paradigm) characteristic, as part of the most advanced and cutting-edge specification book of most real-world monitoring of “human in bed” use-cases, for all relevant detection tasks, we favored clustering schemes involving advanced forms of graph convolutional neural networks. The explainability strongly supports the dependability of the monitoring system, as it provides information of relevance for humans who are in very sensitive and/or dangerous/critical states of health or feeling. A further very sensitive issue for cutting-edge anomaly detection is related to the behavioral evolution over time. Level 1 is the first level at which the first signs of behavioral evolution are detected, although the changes are still relatively small. Each entity of this Level 1 has attributes. Thus, the evolution of the global system behavior may/shall be perceived at this level in one or more of the following ways: (a) soft-evolution, where the value of one or more attributes of one or more Level 1 entities changes over time (slowly or fast); (b) hard-evolution, where a novel Level 1 entity (i.e., not yet encountered until now) happens on the scene; and (c) a mixed-evolution, where both hard-evolution and soft-evolution occur simultaneously.

By carefully reviewing and analyzing various real-world observations of humans in bed in all possible use-cases, we can state that the occurrence of these three forms of behavioral evolution is very likely to be present in the behavior dynamics of the humans to be observed. Consequently, a robust and reliable continual learning capability is required in each advanced intelligent system architecture taking charge of the various Level 1 detection endeavors.

**Consider Level 2**. This is the second level at which relevant behavioral entities are defined, identified and/or detected. At this level, the entities are called SIMPLE EVENTS. Here, we have a second behavior-related aggregation/abstraction of the information provided by the observed Level 1 entities, coded by the sequences over time of those named Level 1 entities. All remarks and observations described for Level 1 w.r.t. the explainability requirement, the learning strategy (unsupervised learning, semi-supervised learning, self-supervised learning, continual learning) and the behavior evolution ways are also valid in Level 2.

**Consider Level 3**. This is the third level at which relevant behavioral entities are defined, identified and/or detected. At this level, the entities are called COMPLEX EVENTS. Here, we have a further behavior-related aggregation/abstraction of the information provided by the observed Level 2 and Level 1 entities, as coded by the sequences over time of those named entities. All remarks and observations described for Level 1 and Level 2 w.r.t. the explainability requirement, the learning strategy (unsupervised learning, semi-supervised learning, self-supervised learning, continual learning) and the behavior evolution ways are also valid at this Level 3.

**Consider Level 4**. This is the fourth level at which relevant behavioral entities are defined, identified and/or detected. At this level, the entities are called BEHAVIOR INSTANCES. Here, we have a further behavior-related aggregation/ abstraction of the information provided by the observed Level 3, Level 2 and Level 1 entities, as coded by the sequences over time of those named entities. All remarks and observations described for Level 1, Level 2 and Level 3 w.r.t.; the explainability requirement; the learning strategy (unsupervised learning; semi-supervised learning; self-supervised learning; continual learning); and the behavior evolution ways are also valid at this Level 4.

### 3.2. Consider the Form-Type Related Anomaly Classification

Most of the anomaly-related literature, especially the ones related to machine systems observations, defines a series of simple and complex forms and types of the anomaly; we can call them “anomaly patterns”. The anomaly concept shall be associated with or linked to a given entity of one of the four highest levels of the hierarchy presented in Figure 3. Indeed, each entity of levels 1 up to 4 (see Figure 3) has a series of attributes that may be hand-crafted by a human designer. Examples of illustrative attributes (just illustrative examples; for a concrete monitoring use-case, the use-case engineer shall determine/define them) of an entity of level 1 up to level 4 (i.e., the levels according to Figure 3):**Attribute 0**: It is a binary attribute indicating whether this entity is new (see novelty detection) or it was already encountered once in the past;**Attribute 1:** Relative occurrence-time difference(s) w.r.t. to one or more of the latest occurrence(s) of the same entity at a given level;**Attribute 2:** Relative occurrence-time difference(s) w.r.t. to one or more of the latest occurrences of other entities of the same given level;**Attribute 3:** Time duration of the current occurrence of the same entity;**Attribute 4:** Selected time-duration-related metadata related to the current and previous occurrences of the same entity;**Attribute 5:** Number of occurrences of the entity with a given time window;**Attribute 6:** Relative number of occurrences of the same entity with a given time window w.r.t. one or more other entities of the same level;**Attribute 7**: Selected metadata w.r.t. one or more of the previous attributes (i.e., Attribute 1 up to Attribute 6);**Etc.**

Some entity attributes may be simple and represented by a single real number/value. However, some other attributes may be complex and represented in the form of vector(s) or matrix/matrices of real and/or binary numbers. Moreover, a coherent normalization of all individual entity attributes values may be of high practical relevance, especially when considering the variety of approaches that may be involved in the anomaly detection endeavor. As one can imagine, an abnormal behavior may be related either to just one or to multiple attributes of a given entity. If one can reliably detect abnormal behavior while involving only a small number of attributes for the individual entities, the better it is. It is also thinkable to merge (this merging may involve an appropriate, eventually complex formula) all several attributes of an entity into one combined, resulting global attribute expressed as a real number. After merging, the combined global attribute (that is, one occurrence in the related time series of its evolution over time), a single real value, may be used as the input for the anomaly detection intelligent scheme.

It is also evident that some of the entity attributes are generated from fully different intelligent processing modules; this is, for example, the case for Attribute 0.

Before discussing the various possible “anomaly patterns”, let us first state that the anomaly assessment may/shall be performed, in parallel, in principle at all four levels of the data-processing hierarchy structure presented in Figure 3. Indeed, the same inherent physical abnormal behavior must be seen at all the five hierarchy levels, starting from Level 0 up to Level 4. One can, however, assume that the related “anomaly scores” (the anomaly score concept is discussed later in the section where the anomaly detection schemes are presented) are not necessarily the same, nor shall the detection be synchronous, respectively, simultaneous at all those data-processing hierarchy levels. 

Table 1 below presents and briefly explains the various forms (i.e., types) of anomalies, which can be observed while analyzing one simple or one global attribute of an entity at any level of the hierarchy presented in Figure 3. In order to be of good use for the practically very important explainability requirement, the anomaly detection scheme/intelligence shall be able to provide the anomaly type and, surely, also the detection confidence level. Without an appropriate posterior adaptation and/or extension, most anomaly detection schemes cannot fulfill these two requirements of providing the specific type of the detected anomaly and the related confidence level. Thus, the anomaly detection shall indicate the entity concerned and its level and the anomaly type along with its related confidence level.

### 3.3. Consider the Observation-Context Perspective Related Anomaly Classification

In this sub-section, we argued that the correct understanding of what an anomaly is strongly depends on the specific observation context perspective. Further, the simultaneous observation of the system from different perspectives also provides a form of theoretical information fusion, which significantly increases the confidence of the anomaly detection process in case a true abnormal behavior is effectively occurring.

The wording “anomaly observation context perspective” refers to the specific element that carries the anomaly label. Indeed, the anomaly predicate shall always be linked to a particular element of the scene or of the system under observation.

The following observation context perspectives (OCP) appear to be of high relevance for the “monitoring of human in bed” general use-case:**OCP 1**: “Condition-type” or better “entity type”-related anomaly. See the entities of the four upper levels of the hierarchy presented in Figure 3. Each of those entities at different levels does define a particular system condition. Indeed, every single time, the system state is represented by a particular entity at each of the four upper hierarchy levels in Figure 3. The condition may be current, past or future; see OCP 5;**OCP 2**: “Conditions-sequence”-related anomaly. The conditions-sequence may be current, past or future; see OCP 5;**OCP 3**: “Single time-window”-related anomaly. Notice that the time window may be current, future or past (see OCP 5);**OCP 4**: “Multiple time-windows”-related anomaly. Notice that some of these time windows may be current, future or past (see OCP 5).;**OCP 5**: “Time-Case“ perspective of anomaly detection. The “Time-Case” concept was defined in Section 1 above. Indeed, this is an additional context feature to be combined with each of the four previous observation context perspectives. Let us recall here those in Section 1 already explained three time cases: (a) Time case 1, the posterior anomaly detection; (b) Time case 2, the online or simultaneous anomaly detection; (c) Time case 3, the forecasting or anterior anomaly detection.

Consider the “Condition-type”-related anomaly perception (OCP 1). In this case, the anomaly is linked to an entity of a given level (see Figure 3). As explained in Section 2, a given occurrence of an entity of a given hierarchy level carries the label abnormal whenever one or several of its attributes (as comprehensively explained in Section 2) present one of the abnormality patterns presented in Table 1. To provide an illustrative example related to a human in bed, “standing-up from the bed” (let us call it entity SB) may be seen as one entity belonging to Level 2 (level 2 is the level of simple events) of Figure 3. Thus, at a particular time, the system state at Level 2 may be represented by this entity named SB and thereby have the label “anomalous” (this depending on the attributes related to it at that time).

Consider the “Conditions-sequence”-related anomaly perception (OCP 2). In this case, the anomaly is linked to an observed sequence of conditions. Here, there is an abnormal behavior if one or more attributes related to this sequence of the condition presents one of the abnormality patterns presented in Table 1. Here, it is a given sequence of conditions at a given level in Figure 3 that obtains the label “anomalous”. Let us define a sequence involving the above-defined entity SB of level 2. A sequence example may be to consider the time between the occurrences of two SB events (immediately consecutive or not) at Level 2 of Figure 3.

Consider the “Single time-window”-related anomaly perception (OCP 3). In this case, the anomaly is linked to a specific time window under observation. The attributes of an observation time window are related to those of the entities at all four hierarchy levels (see Figure 3), which are present therein. Hence, from this perspective, there is abnormal behavior whenever one or more attributes of the target “observation time window” present one of the abnormality patterns presented in Table 1. The length of the time window may be fixed by the monitoring use-case engineer. Here, to involve the above-defined level 2 entity SB, an illustrative example of this use-case may be the consideration of the number and respective time spacings of SB entities within a given “time-window”. Time window examples include the last 6 h, the next 6 h, the last 12 h, the next 24 h, etc.

Consider the “Multiple time-window”-related anomaly perception (OCP 4). This case is very similar to OCP 3 above. The only difference is that more than one time window is observed simultaneously. One or more of these time windows may lie in either past or future or be current.

Consider the “Time-Case”-related anomaly perception (OCP 5). As already explained above, this perspective is complementary to one of the previously described ones, i.e., OCP 1 up to OCP 4. As already discussed in Section 1, one has different sub-regions in the time dimension (see also Figure 4). The element to be assessed (see OCP 1 up to OCP 4) regarding abnormality or not is always located in one of the time sub-regions indicated in Figure 4. The different time cases and how they are specifically named are comprehensively presented in Table 2.

### 3.4. Consider the Learning Strategy and Learning Context Related Perspective for Anomaly Perception

In this sub-section, we discussed the various situations or contexts of the state of eventual prior knowledge about the normal system behavior for a given human in bed observation use-case. Some other relevant information from the real-world context may also be of some use.

The following situations (learning strategies) and issues were analyzed in this sub-section:Unsupervised learning and related system behavior identification, i.e., a “blind” approach;Semi-supervised learning and related system behavior identification, i.e., a “gray” approach;Supervised learning and related system behavior identification, i.e., a “white” approach;The subject-dependency concern w.r.t. behavior identification; there may be some interlinking to either “blind” or “white” approaches;The monitoring use-case dependency; there may be some interlinking to either “blind” or “white” approaches;

Consider the “Global Task” to be realized by the intelligent monitoring system. Independently of all the above-listed situations (learning strategies), the core global task is that of identifying four components of the system to be monitored: (a) Subtask A: identifying the entities of the levels 1 to 4 of Figure 3; (b) Subtask B: behavior monitoring through the evolution over time of the attributes (see Section 3.2) of the identified entities; (c) Subtask C: identification of anomalies while considering the use-case relevant perspective out of those presented in Section 3.3 (see the perspectives OCP 1 up to OCP 5); Subtask D: system behavior evolution identification through either the appearance of novel entities or through the disappearance of entities at one or more levels of Figure 3 or through the time evolution of some of the entity attributes at one or more levels of Figure 3; Subtask E: system behavior forecasting (at the levels 1–4 of Figure 3) for one or more of the future sub-regions indicated in Figure 4.

The learning strategies related discussion in this sub-section is related principally, in the first stage, to Subtask A on the one hand, but also, in the second stage, to the other Subtasks. Indeed, the outcome of Subtask A is the basic infrastructure of the anomaly detection intelligent system. Thus, the other Subtasks (B, C, D and E), which are solved by very advanced models and algorithms, use and are built around attributes and properties of the basic bricks coming from Subtask A.

Consider the situation “Unsupervised learning and related system behavior identification (i.e., a “blind” approach)”. Here, one assumes that no sufficient explicit knowledge is available to identify the entities of levels 1 to 4 of Figure 3. The straightforward and naïve way to identify the behavior of the system under observation is to involve unsupervised learning to identify all entities of levels 1 to 4 of Figure 3 (see Subtask A). The literature is full of approaches capable of reliably performing unsupervised learning; some examples, just to name a few: hidden Markov models, self-organizing maps, various versions of neural graph networks, etc. From the deep-learning literature, a variety of unsupervised learning methods can be found: deep belief networks (DBNs), generative adversarial networks (GANs) [29], variational autoencoders (VAEs), denoising autoencoders (DAEs) and adversarial autoencoders (AAEs) [30]. A particular challenge is, however, to handle the fact that in the context of monitoring a human in bed, the system behavior changes over time. This evolution may result in the appearance of new entities at one of levels 1 to 4 of Figure 3 (see Subtask D). A further issue is that the human samples that may be monitored may have various fully different behavior patterns (example: a 90-year-old lady in a coma; a pregnant woman; a fit, athletic young man who has just been injured in a car accident, etc.). The unsupervised learning performance shall always be the highest possible despite the variety of behaviors amongst the possible human samples to be monitored. The unsupervised learning is real-world realistic and needed, as one generally lack precise information about the specific entities (at the four levels of Figure 3) of a given human in bed sample for monitoring. 

Consider the situation “Semi-supervised learning and related system behavior identification (i.e., a “gray” approach). In some real-world monitoring use-cases, some reliable or reference information may be available regarding specific entities of levels 1 to 4. These specific entities may be the ones of special high interest for the given monitoring use-case. Examples of entities, just for illustration, which may be of interest are: from a sitting position on the bed, the human stands up and leaves the bed; the human is sitting on the bed; the human turns over in the bed, etc. The known entities of special interest may be identified through supervised learning, whereby the other ones, at all hierarchy levels, are identified, for a given human in bed observation sample, through unsupervised learning. Therefore, the overall learning process becomes semi-supervised. Moreover, the unknown behavior evolution always imposes that part of the learning process remains unsupervised.

Consider the situation “Supervised learning and related system behavior identification (i.e., a “white” approach). As discussed above, for semi-supervised learning, fully supervised learning is not realistic for the case of the monitoring of humans in bed. Because a significant behavior evolution can always be assumed, a fully supervised learning strategy is not appropriate. However, self-supervision is welcome and can support the system behavior evolution. 

Consider the situation “The subject-dependency concern w.r.t. behavior identification”. Subject-dependency is a very sensitive issue in human-related monitoring systems, which may be intrusive or non-intrusive. 

It is well-known from the broad research involving the machine learning-based observations of the dynamics of human beings in various use-cases and applications such as “emotion detection”, “stress detection”, “pain detection” and “activity monitoring/classification” that there is a significant subject-dependency in the data patterns. For example, just for illustration, the data patterns corresponding to a given emotion, such as “joy” for a subject/person A, may be fully/very different from the data pattern corresponding to the same emotion, “joy”, for a different subject/person B. For our core use-case of monitoring a human in bed, one activity, “standing-up to leave the bed”, may produce data patterns that may be significantly dissimilar depending on the specific subject under observation (see Figure 2). Some objective reasons explaining the difference in those patterns may be a difference in age, difference in gender, difference in weight, difference in health conditions, difference in mood, difference in fitness status, difference in race/tribe, etc.

An interesting question for the related research is however to holistically comprehend whether there is some significant subject-dependency of relevance for the learning processes (either supervised or unsupervised or semi-supervised) to ensure the reliable detection of anomalies in our target core context of monitoring a human in bed. A particularly interesting issue is whether data to be used for training and/or testing can be shared amongst subjects or not. Such a sharing is meaningful for models involving neural networks and/or deep-learning while applying the so-called transfer learning [31,32,33]. Another issue is how far subject-dependency can be taken into account in the endeavors related “data augmentation” [34] and “artificial data-set generation” [35]. It is well known in the relevant scientific community that these two last-named endeavors are very important for the development and tuning of truly robust models (for anomaly detection and forecasting), but also for the stress-testing, the verification and the validation of developed robust and adaptive abnormality detection models. We discussed these concerns more deeply in a sub-section (see Section 6) further below.

Consider the situation “The monitoring use-case dependency”. Some aspects or requirements of a given monitoring use-case may be relevant for or have a significant influence on the learning process setting. Indeed, each of the humans to be monitored may have their own specific issues. Additionally, the core monitoring objective may be very specific for each case. Some illustrative examples of “use-case specific perspective (USP)” may be given for a better understanding: (a) For a given client under monitoring, only one or more entities of the levels 1 to 4 of Figure 3 are of interest; (b) For a given client, one has a clear expectation of the evolution of one or more behavioral patterns; (c) For a given client, only the prediction of the “occurrence time” of a particular event or of specific behavior to be either below or higher than a fixed threshold is what is of high interest; (d) For a given client, it is the absence of a given event or of a specific behavior over a fixed time period which is of interest (this time period may cover or not a part of the past, a part of the current times and a part of the future time), etc. Moreover, the same client may undergo at different consecutive times two different of these so-called USPs. For example, a person is sick (they had a stroke) and is being treated for a couple of weeks; in this period, there may be a particular clear/known expectation w.r.t. behavior, a given USP. Then he recovers, but he must also be further observed for a certain time of some weeks; for this second observation period, the expectations are (or may be) fully different and expressed by a different USP. In general, one may have USP-specific observations but also fully neutral observations, which detect abnormalities fully blindly. These few examples suggest that the background and the overall prior knowledge of the use-case engineer may be of very significant relevance for the training process settings and even for the definition of the above described OCPs (observation context perspectives). One also understands that the use-case engineering may fix or determine which, amongst the various anomaly patterns presented in Table 1, are highly relevant and interesting for a given subject monitoring case. From practical regard, if the same intelligent system must be reconfigured to fit the use-case nuances/objectives/specifics of different patients to be monitored on the same or different beds, the overall systems engineering must seriously find an appropriate way to integrate the “subject and use-case”-specific complex reconfigurability capacity of the “anomaly detection intelligent system”. The overall requirements engineering shall therefore decide whether a USP neutral and universal abnormality detection is wished or whether one rather needs a detection system that can be dynamically reconfigured by the system operators depending on various USP they may have. Essentially, the USPs can be viewed as special augmentations of the OCPs.

### 3.5. Consider the “Novelty Related Anomaly” Perception

According to the literature [36], novelty detection can be defined as the task of recognizing that the data being observed now differ in some respect from the data that were observed until now (e.g., during training). Its practical importance and challenging nature led to many approaches being proposed in the literature.

Indeed, the last decades witnessed significant developments related to the need for self-adaptive and self-organizing systems capabilities. A fundamental ingredient in such systems is the ability to recognize unanticipated and dynamic conditions, which require robust autonomous learning capabilities. A common fundamental assumption, which is valid for the monitoring of a human in bed, is that the distributions of normal data (at all five levels of Figure 3) are eventually non-stationary and time-variant. This means that they change (eventually significantly) over time. This property is traditionally called “concept-drift”; it expresses a “behavioral change”, which is a very sensitive issue.

It is very crucial to properly and carefully differentiate and feel the sensitive nuance between “anomaly” and “novelty” from a very general and holistic perspective [37]. The traditional definition of novelty from the machine learning literature is too vague as it states that “novelty detection is the task to identify observations that differ from previous experiences or expectations”. We rather prefer and agree with the other statement from the related literature that reads as follows: one cites, “one shall refer to novelties as agglomerations of coherent anomalies (i.e., abnormal observations), representing a fundamental change in the underlying processes generating the observations”. A deep analysis of our suggested structuring of possible occurrences of anomalies in Table 1 suggests that some of those defined anomaly patterns are already usable or relevant for a merging with some forms of novelty patterns (see AN-4 to AN-9). A later deeper analysis of the various novelty patterns sheds more light in this respect. Overall, outliers (but also all anomaly patterns AN-1 to AN-3 in Table 1) are considered to be isolated non-systematic deviations from the “normal” behavior. A novelty can be practically viewed/assessed by means of a deviation or a distance between a set of current observations forming a model of the (past/historical) perception (i.e., generated at startup time or in previous observation cycles) and a model of the expected (now/future) observations.

It is important to differentiate clearly between “anomaly” and “novelty”. These two concepts may partially overlap. Indeed, the “anomaly” concept (and its related detection) is related to an independent observation at a given time point. In contrast, the “novelty” concept is located at a conceptually higher level, whereby one observes patterns of successive occurrences of “anomaly” observations. These patterns may show structures, one of them being agglomerations or bursters of “anomalies” in the form of clusters. Thus, the history of observed and/or predicted anomalies or anomaly pattern is the essential input of the novelty detection functionality. A part of the anomaly patterns described in Table 1 is already very supportive of the novelty identification functionality (see AN-4 to AN-9). Another strong support of the novelty modeling and identification provided in/by our overall here suggested ontological framework is the various OCPs (anomaly-related “observation context perspectives”), which are comprehensively presented in Section 3.3.

It is thus clear and understood that the systematic occurrence of multiple anomalies is usually a reliable indicator for a significant change that requires a model adaptation. Novelty detection is, therefore, essentially and precisely the detection of such systematics of multiple anomalies. It is therefore very good that our multidimensional ontological context structuring w.r.t. anomaly identification (see OCP 1 to OCP 5 in Section 3.3) lays a solid ground on which a robust “novelty detection” functionality can reliably operate.

Some sources in the literature look at the novelty phenomenon just from a “temporal perspective” (see Table 3, which is not even yet exhaustive). In order to be holistic and more comprehensive, we suggest that the novelty detection/analysis be based on the more comprehensive anomaly perception involving all five reference OCPs described in Section 3.3; one shall even additionally integrate the so-called USPs (use-case specific perspectives; see Section 3.4), if any, in order to ensure that all practical concerns of relevance are effectively taken into account. The perspective described in Table 3 is, therefore, de facto significantly extended by the considerations of both the OCPs and the USPs. 

### 3.6. Subject-Dependency Related Anomaly Perception, Detection and Related Nuances

The general framework formulated in this paper is such that the general architecture presented in Figure 3 is valid for any human subject to be considered for observation in a bed. There are, however, differences (from small to big) in instances of the architecture of Figure 3 between two given different subjects that are related to the following aspects:(a)The number of entities at each of the 4 four upper levels of Figure 3;(b)The specific patterns of each entity at all four upper levels of Figure 3;(c)The attributes’ values related to each entity at all four upper levels of Figure 3.

Beyond these three listed differences, which are subject-dependent, all other dimensions and perspectives of the “abnormality” concept (which are extensively presented and discussed in this paper) are independent of a given subject and are universally valid for all.

Our core focus here is the context monitoring of humans in bed. However, humans may behave very differently regarding their physical movements in the bed. A small child, a pregnant woman or a 95-year-old lady do not behave the same while lying in a bed. Moreover, their respective health status and/or dynamics may impact their physical movements in the bed. This is the first important difference between a human and a machine. Machines of the same type are expected to display the same nominal behavior. However, this is not the case for humans. Even two humans with profiles close to each other (e.g., same age, same weight, same health status, same gender, etc.) may nevertheless behave fully differently regarding their physical activity-related behavior(s) in the bed.

The first monitoring intention for a human in bed is essential to classify instants (or time windows) of subject-dependent normal and/or regular behavior and instants (or time windows) of abnormal behavior. Here, the “normality” perception is specific and related to each individual human sample under monitoring. An intelligent model trained with data from one given human sample does not perform well if tested on a different human sample; abnormality will/would be detected at all levels of Figure 4. Nevertheless, subject-dependently trained models may be used (if needed) for identification purposes of individuals known a priori.

Furthermore, the data archive used for models trained on a huge number of different human samples may be later used for a comprehensive clustering of the human samples (i.e., human population) observed so far. Such clustering may lead to the definition of a reference dataset and/or reference users representing each of the fixed/defined clusters (of human samples). This is very meaningful in the current area of big data. Moreover, for benchmarking and validation purposes of different models that can be developed for abnormality detection and forecasting, such reference datasets (and users) for the various clusters are a very precious resource. Moreover, for better training of the various models under development, especially those involving neural networks and deep learning, those reference datasets can also be used to construct generative models to be used for extensive and reliable data augmentation endeavors through the generation of artificial data, which are nevertheless sufficiently reflecting the real-world conditions of real humans. Generative adversarial models trained on real field data can be used to generate more and still very realistic artificial data [38,39]. Such artificial data have some interesting features when compared to real data collected from real human samples: (a) they are anonymous, (b) they can be available in a huge quantity and (c) they also fill the gap of providing a reliable dataset to be used for comprehensive system verification and system validation endeavors of intelligent models developed for anomaly detection and/or forecasting in the case of monitoring human in bed.

Due to the technical possibilities of transfer learning [40], a model training over data from the various above-mentioned possible human sample clusters shall also potentially lead to more accurate and more robust models for both detecting and forecasting normal/abnormal behavior.

A particularly innovative modeling concept/framework may be based on the extensive data mining of the extensive reference datasets mentioned above, which may include, in addition to the data collected from a very high number of human samples that are fully representative of the target population of humans, a significant portion of artificially generated but still realistic reference data. 

Hereby, one can define one or more “reference time lengths” for BEHAVIOR SEGMENTS. For example, for illustration: (a) reference behavior length of 1 h; (b) reference behavior length of 6 h; (c) reference behavior length of 12 h; (d) reference behavior length of 1 day; (e) reference behavior length of 3 days; (f) reference behavior length of 7 days.

After the different reference time lengths have bee defined, extensive data mining can be performed over the huge above-mentioned reference datasets. Thereby, for each reference behavior time length, a clustering of the respective behavior segments (contained within the huge reference dataset) is performed. One then obtains the following situations, presented in this form for illustration purposes:(a)**Reference Clusters** for behavior segments of length 1 h: each cluster shall be considered as a class, examples: 1HC1, 1HC2, 1HC3; …; 1HCN_1_. Hereby, N_1_ is the total number of classes of length 1 h;(b)**Reference Clusters** for behavior segments of length 6 h: each cluster shall be considered as a class, examples: 6HC1, 6HC2, 6HC3; …; 6HCN_2_. Hereby, N_2_ is the total number of classes of length 6 h;(c)**Reference Clusters** for behavior segments of length 12 h: each cluster shall be considered as a class, examples: 12HC1, 12HC2, 12HC3; …; 12HCN_3_. Hereby, N_3_ is the total number of classes of length 12 h;(d)**Reference Clusters** for behavior segments of length 1 day: each cluster shall be considered as a class, examples: 1DC1, 1DC2, 1DC3; …; 1DCN_4_. Hereby, N_4_ is the total number of classes of length 1 day;(e)**Reference Clusters** for behavior segments of length 3 days: each cluster shall be considered as a class, examples: 3DC1, 3DC2, 3DC3; …; 3DCN_5_. Hereby, N_5_ is the total number of classes of length 3 days;(f)**Reference Clusters** for behavior segments of length 7 days: each cluster shall be considered as a class, examples: 7DC1, 7DC2, 7DC3; …; 7DCN_6_. Hereby, N_6_ is the total number of classes of length 7 days.(g)**Etc.**

The clustering (of the different behavior segment lengths) may involve just one level (e.g., just Level 1 or just Level *n* (*n* = 1, 2, 3, 4)) or more levels of Figure 3. 

For each reference behavior time length, there is a given number of reference behavior clusters. Each of these clusters may be called a “behavior class” of the respective length. Thus, the behavior of a given human under observation may be described by the sequence of behavior classes of the different reference lengths. In this way, the ontology for a universal behavior description would be that of indicating the sequence of observed and/or predicted behavior classes (of given reference lengths, of uniform or mixed lengths) for a given human under observation. One can view the reference length as a form of observation granularity or observation resolution. In a given observation case, one has several choices w.r.t. the observation resolution, for example, just to name a few: (a) consider a sequence of one single resolution; (b) consider simultaneously two sequences of different resolutions in parallel; (c) consider a sequence of mixed resolutions.

These defined behavior classes are essentially just a form of the universal alphabet (this alphabet is fully user-independent) for universally describing the behavior of a given individual, “human in bed”, under observation. The reference clusters (i.e., classes) defined above provide a universal subject-independent naming of the entities that can be observed at all four levels of the architecture presented in Figure 3. This universal alphabet fits well in our present area of big data.

All the perspectives defined in this paper for the understanding of the anomaly concept can now be linked and/or adapted to this new alphabet of behavior segment classes. As already discussed above, this new alphabet requires a good reference dataset (ideally open data), which shall be accepted and considered as a standard for all. Thus, the subject-dependent anomaly detection and/or forecasting can now be learned and performed according to the different OCPs (observation case perspective) described in Section 3.3.

## 4. Comprehensive Discussions of Selected Key Advanced Characteristics of Robust and Real-World Mature Anomaly Detection Schemes

The overall context of this study and review is the monitoring of the physical activity of human beings in bed. The system (or samples of it) to be monitored has some of the most difficult system modeling-related characteristics, namely: (a) two different samples of the system do not behave similarly; (b) the system dynamics display a significant stochastic flavor; (c) the system behavior is time-variant; (d) the observation time horizons, in which abnormality related warning(s) is/are expected, may vary from short-term (some minutes to some hours), middle-term (some days) to long-term (several weeks); (e) the knowledge about the future evolution of the system behavior may be involved in the assessment of the current status; (f) because of the system evolution, a previously rated abnormal behavior may be at a posterior time viewed normal and vice-versa; (g) a certain tolerance level/grade, which may be different due to the situation of different human samples, can be relevant for the abnormality assessment; (h) in several cases, the abnormality information has a significant meaning/implication w.r.t. the health/medical status of the given system sample, this makes the system samples to be observed to be ranked/assimilated into the so-called (in this case “soft”) “safety”-critical systems; (i) amongst others because of (soft) safety-criticality the abnormality detection must also include a related uncertainty information or confidence level; (j) amongst others because of the (soft) safety-criticality, an explainability of both the abnormality assessments/predictions and the system evolution’s detections is needed and very useful for most application scenarios.

Because of the complexity of the system (and the related diversity amongst the system samples) to be monitored (activities of a human in bed) and the above-described characteristics, a mature monitoring system integrates the functionalities described in Table 4.

## 5. Comprehensive Requirements Engineering Dossier for a Robust Anomaly Detection System

Essentially, Section 4 already provided the quintessence of the truly very tough requirements dossier for a modern advanced monitoring system of a human in bed w.r.t. abnormal behavior identification and forecasting (see Table 4). It is evident that this requirements dossier reaches far beyond the current state-of-the-art, although major model bricks that can be used to satisfy all of the requirements listed in Table 4 are already available in the literature [41,42,43,44,45]. Nevertheless, an appropriate tuning, adaptation and integration of those model bricks are necessary.

For a given observation scenario, the use-case engineer defines or fixes the appropriate USP and OCP configurations. Regarding the overall anomaly detection system(s) w.r.t. taking into account the practical needs of the real-world scenarios, almost all the hard requirements (see REQ 1 to REQ 11) described in Table 4 must be fully or partially satisfied.

The comprehensive requirements dossier presented in Table 4 is a first conceptual benchmarking of all anomaly detection schemes developed for human activity monitoring. This also has significant implications for (a) the verification and validation of human activity monitoring schemes and (b) the critical review of the relevant literature (related works) w.r.t., establishing how far the various relevant anomaly detection-and-prediction models and systems fulfill or do not fulfill the requirements presented in Table 4.

## 6. Comprehensive Verification and Validation Challenge for a Robust Anomaly Detection System

“System verification” is an important part of the overall systems engineering process. Essentially, it checks the correctness of either a complete system or a system sub-component or both. This covers the complete life cycle of any technical system.

More precisely, verification is the process for determining whether or not a product (here, it is the anomaly detection-and-prediction system) fulfills the requirements or specifications established for it (here, the requirements are presented in Table 4). On the other hand, “System Validation” is the assessment of a planned or delivered system to meet the sponsor’s operational needs in the most realistic environment achievable (here, the real-world setting or real field data must be involved), which is also very important.

For both verification and validation, appropriate data must be available, which are good and/or satisfactory w.r.t. quality and quantity. Since field data are generally very expensive to collect, a very pragmatic question is whether usable data (having a minimum required quality) can be generated artificially. Such a generation is, therefore, a form of dependable “data augmentation” strategy. This data augmentation, especially for the verification exercise, provides more data samples (each sample representing a different human under observation) covering a longer time horizon (each data sample is much longer regarding time). In a later subsection, we suggested that a series of generative adversarial models, especially from the deep-learning literature, already have a significant maturity level to enable the generation of an extensive, dependable artificial dataset that can then be used for comprehensive verification. Regarding the validation, after a very comprehensive verification, solely real-world field data must be involved.

Regarding both verification and validation, however, good and appropriate performance metrics are needed. Some interesting metrics were suggested in the relevant literature [46]. Before one defines better ones or additional ones, these metrics are briefly described and commented on in Table 5.

## 7. Modelling the Basic Infrastructure of the “Monitoring System of Humans-in-Bed” in the form of a Graph Network

The general structure and infrastructure of the intelligent monitoring system presented in Figure 3 can be (and shall be) represented in the form of a “Graph Network”. Because of the time evolution, a so-called “Dynamic Graph Network” is more appropriate. As is explained below, using graph networks for modeling the complex system at stake has a lot of unique advantages [47], especially while considering the very tough requirements presented in Table 4. The graph network representing the scene of Figure 3 is thus a good platform on top of which the various intelligent processing bricks are integrated in the effort to satisfy most (if not all) of the requirements described in Table 4. The review paper of Xiaoxiao Ma et al. (2021) [47] provides a more than sufficient review of the potential of dynamic graph networks for modeling a given monitoring scene for the purpose of robust abnormality detection. We just need a careful mapping of all elements of the architecture of Figure 3 into a dynamic graph network, which integrates/models all relevant perspectives and issues: OCPs, USPs, entities—attributes (see levels 1–4), Figure 4, etc. 

Consider the dynamic graph network representing the scene, see Figure 3. Here, the instances (or entities) of layers 1 to 4 of Figure 3 are represented by nodes of the dynamic graph network. Additionally, the so-called attributes of those instances, see Section 3.2 (Attribute 1 to Attribute 7, etc.), are represented in attributes of edges between relevant nodes. Moreover, the so-called OCPs and USPs (see Section 3.3 and Section 3.4) can be integrated (modeled appropriately) into the attributes of edges between relevant nodes.

In Table 6, we summarized a series of hard reasons pleading for the use of graph networks for comprehensive modeling of the scene presented in Figure 3 in the perspective of abnormal behavior detection while considering, additionally, the OCPs, USPs, Figure 4 and Table 2 and Table 4.

While using the framework of a graph network for modeling objects or entities, these can have one of two possible anomaly-related attributes: (a) the object or the relation is “standard”, “normal” or “expected”, and (b) the object or the relation is “abnormal”. The traditional schemes, which represent real-world objects as feature vectors, are incapable of enabling methods capable of detecting the inherently complex relationship(s) between objects. The incapability to grasp and detect the relationships between objects is a severe limitation. Because in our target context of “monitoring a human in bed” (see Figure 3 and the OCPs and USPs), these complex relationships are very important. Indeed, in real-world contexts, beyond the standard attributes related to an object itself, eventually various and rich relationships between objects belonging to the same context also exist. These various relationships provide additional and/or complementary information, which supports the robust detection of anomalies related to either objects/entities or rather to the relations between objects. In the graph-based modeling paradigm, nodes/vertices relate to real-world objects. Furthermore, edges relate to relationships between objects/entities. Moreover, sub-graphs, which include both nodes and edges, are another level of entities (i.e., complex entities) that can also bear, at a given time, the attribute of being either “normal” or “abnormal”. 

For illustrative purposes, Figure 5 highlights the essential difference between conventional anomaly detection and graph-based anomaly detection. One can see that the graph-based anomaly detection detects anomalies in a much more complex, broader and richer ontological framework: object anomaly (node anomaly), relation anomaly (edge anomaly), group of objects related anomaly, sub-graph related anomaly, etc.

Graph based anomaly detection does differentiate at least three types of anomalies:(a)**Node-related anomalies**, which can be either local or global. A node that is labeled globally abnormal is thereby compared to all other nodes of the whole graph. However, a node may also be locally abnormal when it is compared to the other nodes belonging to a local sub-graph (around a given node) of the whole graph;(b)**Structure-related anomalies**. Structural information focuses on the relationships amongst objects/entities, which are represented by the edges connecting the nodes in the whole graph. Here, the abnormality is related to the different connection patterns. Evidently, the abnormality of an edge does also affect the involved nodes;(c)**Community anomalies**. Here, one considers a sub-graph of the whole graph that is called abnormal. Hereby, information related to both node attributes and edge attributes is involved in the assessment of the status “abnormal” for a given sub-graph.

These graph-based anomaly perspectives complement/extend the anomaly classification presented in Table 1 and Table 2.

Due to the very significant good characteristics and features of a graph network-based scene modeling, we pleaded, for our target context of “monitoring of a human in bed”, that the comprehensive global anomaly detection intelligent system be based on a dynamic graph network paradigm. Essentially, it shall be (mapping Figure 3 at the basis) a complex “Attributed Dynamic Graph”. Details are provided in Section 9 below. There are a series of interesting works that provide useful basics on how to represent a dynamic scenario, such as the one presented in Figure 3 (including the OCPs and USPs) through a dynamic graph, just to name a few; e.g., [49]. Indeed, it is well known that learning graph representations are a fundamental task aimed at capturing various properties of graphs in the vector space. The real-world system/network presented in Figure 3 (when considering OCPs and USPs) evolves over time and has undoubtedly varying complex dynamics. Ensuring a good capturing of such an evolution is key to predicting the properties of unseen networks. In order to understand how the network dynamics affect the prediction performance, one must design an appropriate embedding approach (one example from the recent literature is “dyngraph2vec”, which may be extended/adapted to fit the appropriate mapping of the scene presented in Figure 3; other examples from recent literature are “dynamicTriad” [50] and “dynGEM” [51]) that learns the structure of the evolution in the underlying dynamic graphs and can then predict unseen links and nodes (and their respective attributes) with higher precision.

After a scene modeling through a dynamic graph network is performed, now given the observed temporal snapshots of graphs (at some consecutive discrete times), a key goal is then to learn a representation of nodes and edges at each time step while capturing the dynamics such that one can then predict their future configurations or structural patterns (nodes, edges and related attributes). Learning such representations is a knowingly challenging task. Indeed, a dynamic graph embedding, which considers multiple snapshots (at consecutive discrete times) of a graph and obtains a time series of vectors for each node/edge, is highly needed and very useful.

On top of a dynamic graph embedding (see Goal et al. (2020) [49]), a merging of both pattern mining and feature learning can be integrated to realize a robust graph-based anomaly detection as suggested for example by Zhao et al. (2021) [52], Ma et al. (2021) [47], etc.

## 8. Comprehensive Critical State-of-the-Art Review of Core Approaches Usable to Construct a Robust Anomaly Detection System

The broad relevant literature on anomaly detection in general, including the works related to human activity monitoring in particular, can be divided into three major avenues or approaches (abbreviation used later for “major avenue/approach: MAJA) and/or a combination/merging of some or all of them. In this section, each of these major approaches (MAJA) is briefly presented/described, and a couple of related representatives’ most recent works/papers are referenced. Indeed, a very rich literature on each of the MAJAs exists, and even very useful, very comprehensive related survey papers. The critical qualitative critical review consists of checking and assessing how far the quintessence of these major avenues is potentially capable of satisfying (fully or at least partially) the particularly hard requirements presented in Table 4. For each requirement of Table 4, the discussions and statements related to each of the MAJAs shall be underscored by one or more representative published works. 

### 8.1. Brief Description of the Quintessence of the Four Major Usable Approaches/Avenues/Paradigms (i.e., Anomaly Detection Paradigms) for Human Activity Monitoring

A careful survey of the relevant and most recent literature reveals the following major approaches or paradigms for anomaly detection and/or forecasting, which are also an application for the target context of “monitoring the physical activity of a human in bed”:MAJA 1: Statistical- or “stochastic processes”-based methods;MAJA 2: Deep-learning- and neural network-based methods;MAJA 3: Graph-network-based methods, combined with either traditional ML (machine learning) methods or MAJA 1;MAJA 4: Graph neural network (or Graph-based DL) based methods.

The quintessence of each of these lastly listed MAJAs is comprehensively presented in Table 7. It gives a sufficiently comprehensive global view (i.e., a big picture) of all those different paradigms. The described respective core quintessence of each of the approaches identifies both the potential and limitations of each of the major approaches.

### 8.2. Assessment of How Far the Four Major Approaches/Avenues Do Fulfil or Not the Comprehensive Real-World Requirements Formulated in Section 5 (see Table 4)

In this section, we carefully analyzed/assessed the four major approaches presented in Table 7 w.r.t. their capabilities to fully or partially satisfy the challenging requirements dossier presented in Table 4. The comprehensive outcome of this deep and careful assessment is summarized in Table 8. As one can see from Table 8, MAJA 4 (that is, the graph DL approach) is unequivocally the approach with the truly highest and best potential for fully satisfying the hard requirement dossier, which takes all real-world and practical needs into consideration, especially for the target context of “monitoring a human in bed”. Overall, DL and graph involving approaches appear to have more potential to tackle the tough real-world requirements. MAJA 4 is the frontier (that is, cutting-edge) approach that offers the maximum flexibility and reliability to satisfy the complete requirements dossier expressed in Table 4 fully.

It is further clear that a monitoring system of humans in bed, which satisfies the specifications described in Table 4 and Table 8, if extensively and comprehensively verified and validated, is a frontier system in the market and shall enjoy a high acceptance by the operators.

## 9. Discussion of a Tentative General Strategic System Architecture Potentially Satisfying the Formulated Hard Requirements

As comprehensively and extensively discussed in Section 8, a system based on Graph-DL (MAJA 4) is the one that has the potential to reliably solve the tough endeavor expressed in Table 4 for the monitoring of humans in bed. The MAJA 4-based system is very innovative as it is also a truly explainable, interpretable and dynamically reconfigurable anomaly detection system. Beyond the target context of monitoring humans in bed, this system is applicable to all other human activity monitoring contexts and use-cases eventually without exception because of the very strong and flexible expressive modeling graph-based neural networks. It has indeed very superior characteristics that evidently reach far beyond the current state-of-the-art.

Since “Graph-DL/MAJA-4” is very flexible and applicable to all human activity monitoring contexts and use-cases, its verification and validation can be conducted by using the diverse public datasets, which were used for research and benchmarking purposes by the relevant research community.

From a practical System Engineering perspectives, the development of the Graph-DL/MAJA 4 shall follow, roughly, the following key steps: (1) STEP 1: Use-case survey and use-case engineering after comprehensive survey of all possible application contexts; (2) STEP 2: Based on the insights provided by STEP 1, comprehensively formulate a representative set of practically realistic OCPs and USPs (respective realistic tolerance margin ranges shall also be collected); (3) STEP 3: Comprehensive architecture design, involving Graph-DL of the architecture in Figure 3 while considering the different OCPs, USPs and the various explainability and interpretability and reconfigurability concerns described in Table 4; (4) STEP 4: Architecture tuning, optimization and system verification by using a selection of public datasets for extensive stress-testing; STEP 5: Final validation by involving the observation in real-time, over a period of several weeks, of real human clients of various significantly diverse profiles w.r.t. age, gender, health status, weight, height, special personal issues/situations (e.g., a lady before and after giving birth, a person before and after undergoing a chirurgical/medical intervention, a person after stroke, etc.), etc; STEP 6: User manual formulation for the system operators in the practice w.r.t. to the configuration for a given use-case and also w.r.t. the complex multi-dimensional reconfigurability and explainability characteristics. 

## 10. Conclusions and Outlook

Truly—and the rich literature on the topic underscores it—abnormality detection and/or prediction is a very hot topic in several technical and socio-technical contexts. In this paper, we addressed it in the frame of the activity monitoring of a human in bed. Activity monitoring of humans in bed is very relevant for various applications areas and contexts: medical, non-medical, some forms of leisure, older people, etc. These various application contexts result in a huge variety of nuances w.r.t. the appropriate system configuration needed according to the 15 requirements presented in Table 4.

Essentially, this paper presents a comprehensive formulation of a requirements engineering dossier for the monitoring system of a “human in bed” for abnormal behavior detection and/or forecasting. Hereby, practical and real-world constraints and concerns are identified and taken into consideration in the requirements dossier. A comprehensive and holistic discussion of the anomaly concept is extensively conducted, and it also contributes to laying the ground for a realistic specification book of the anomaly detection system. Some other system engineering relevant issues were briefly addressed, e.g., just to name a few, verification and validation. Of particular interest is the comprehensive and structured critical review of the relevant literature, which leads to identifying four major approaches of interest. These four approaches are comprehensively evaluated from the perspective of the requirements dossier. It is hereby clearly demonstrated that the approach integrating graph networks and advanced deep-learning schemes is the one capable of fully fulfilling the challenging issues expressed in the “real-world conditions aware” specification book. To conclude, a few recommendations regarding system architecture and overall systems engineering were formulated.

A good number of systems close to the market entrance are mostly built around MAJA 1. The P.SYS system is a good representative example of them. They enjoy especially the practically very interesting features of being low-cost and low energy consumption while ensuring a robust/solid data security and a self-sufficient operation. This is already sufficient for a huge number of practical, real-world use-cases and/or scenarios. Nevertheless, future systems integrating MAJA 4 provide more superior system-related features w.r.t. all 15 requirements listed in Table 4.

Indeed, the “Graph-DL/MAJA-4” system concept suggested in this paper is very innovative as it is also a truly explainable, interpretable and dynamically reconfigurable anomaly detection system. Beyond the target context of monitoring humans in bed, this system is applicable to all other human activity monitoring contexts and use-cases without exception. It has indeed very superior characteristics, which evidently reach far beyond the current state-of-the-art.

Future works will focus first on a comprehensive scenario and use-case analysis that considers a sufficiently representative set of realistic application contexts. Then, they shall follow a careful and holistic system engineering process as suggested in Section 8. After a comprehensive validation, a monitoring system implementing the system concept of “Graph-DL/MAJA-4” will be a frontiers (i.e., cutting-edge) system demonstrating a strong superiority compared to competing systems. The market success chances of a company owning such a system are great [1].

## Figures and Tables

**Figure 1 sensors-22-06279-f001:**
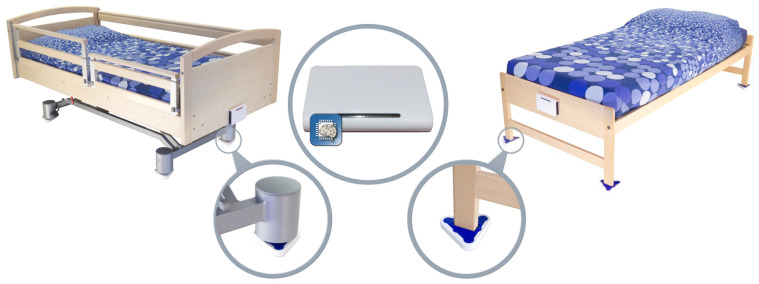
Human in bed monitoring through signals generated by sensors placed under the bed. Under each of the four bed legs, two sensors are placed: one weight sensor and one motion sensor. Thus, eight sensor measurement values are generated continuously over time for further processing by the anomaly detection intelligent system [25].

**Figure 2 sensors-22-06279-f002:**
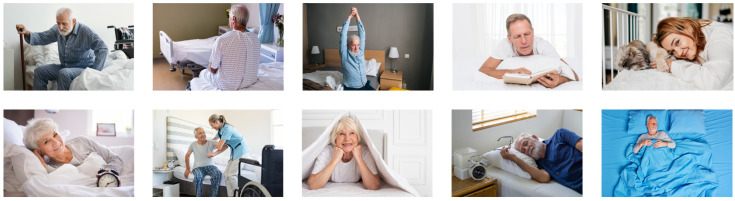
There is a huge variety of possible static and/or dynamical activities of a human in bed, which are monitored through a sensor system such as the one presented in Figure 1. The intelligent system to process the sensor data is capable of detecting and, eventually, also predict normal activities and abnormal ones. (Source of the different image parts: Freepik).

**Figure 3 sensors-22-06279-f003:**
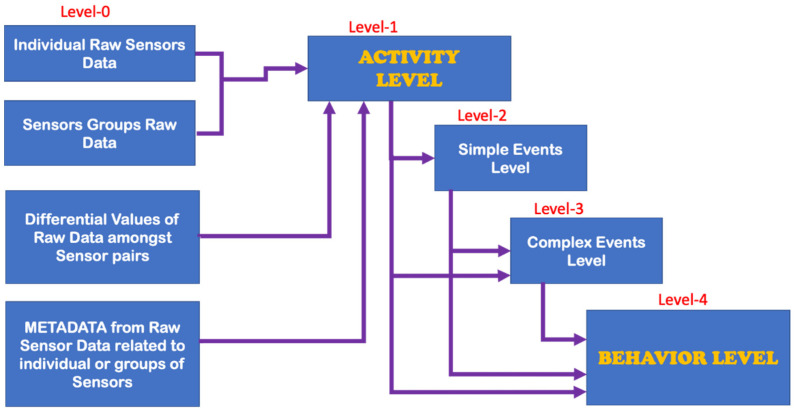
Presentation of the four major data-processing levels that were used in the comprehensive ontological framework for defining the general “anomaly” concept.

**Figure 4 sensors-22-06279-f004:**
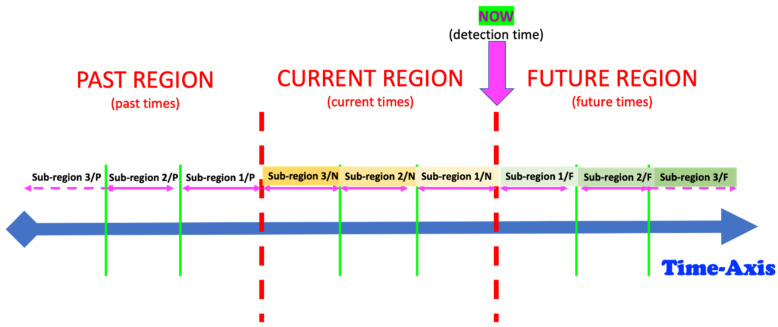
A possible and useful comprehensive structuring of the time dimension into regions and sub-regions. The use-case engineer shall fix meaningful durations/lengths of the different sub-regions.

**Figure 5 sensors-22-06279-f005:**
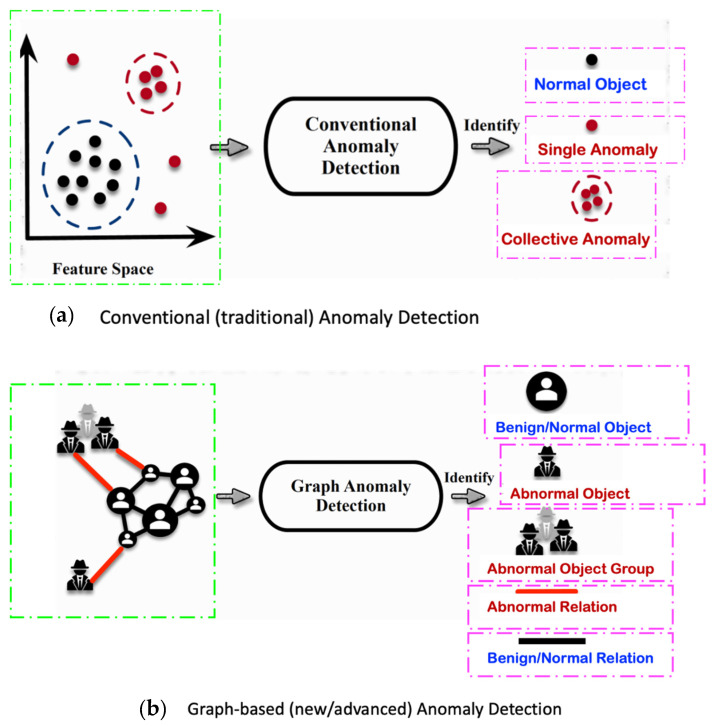
For illustration, Toy Examples—A comparison of “Conventional Anomaly Detection” (see part (**a**)) and “Graph Anomaly Detection” (see part (**b**)). Apart from anomalies shown in part (**b**) of the figure, graph anomaly detection also identifies graph-level anomalies.

**Table 1 sensors-22-06279-t001:** Overview of all anomaly patterns that may be observed/detected w.r.t. a given simple or global attribute of an entity belonging to any of the four highest levels of the data-processing hierarchy of Figure 3.

Anomaly Pattern Name and ID	Point Observation vs. Burst Observation Versus Interval Observation	Related Illustrative Graphical Illustration	Remarks and Other Useful Hints and/or Considerations
Outlier (AN-1)	Point observation	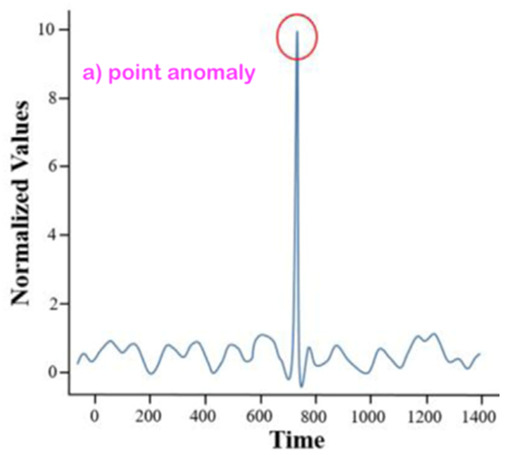	It is generally a stochastically described irregularity. Indeed, probability densities are calculated for target parameters, and defined percentiles are declared as outliers and thereby as anomalies of a certain degree.
Collective anomaly (AN-2)	Burst observation	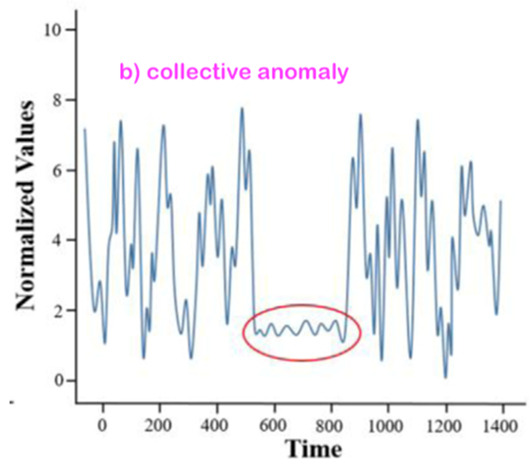	Collective anomalies are data points that are considered anomalies when viewed with other data points against the rest of the data set.
Contextual anomaly (AN-3)	Point observation	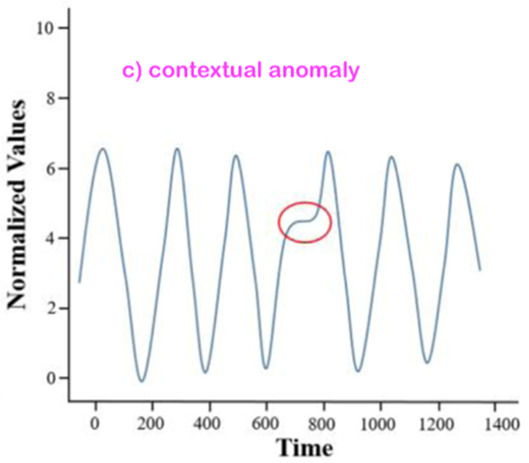	Context anomalies are data points that are considered abnormal when viewed against meta-information associated with the data points.
Missing signal/data anomaly (AN-4)	Time interval observation	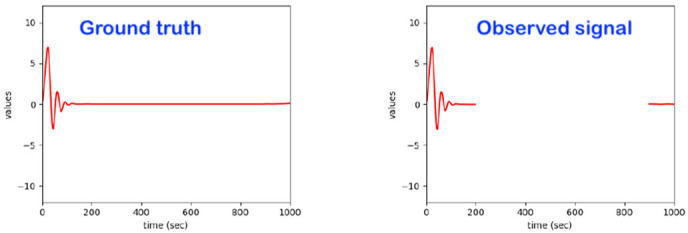	Here, most of the data are missing, and the time/frequency response is zero. This anomaly type is more relevant for Level 0 of Figure 3.
Minor data/ signal (AN-5)	Time interval observation	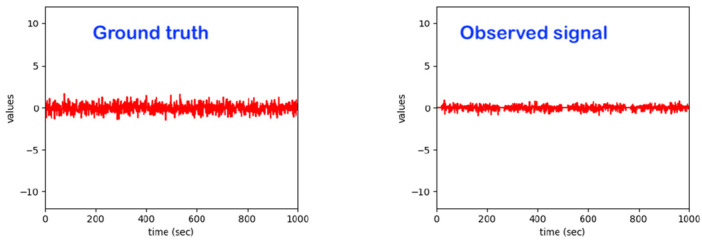	Here, compared to the ground truth values, the observed amplitude is very small.
Multiple outlier pattern (AN-6)	Time interval observation	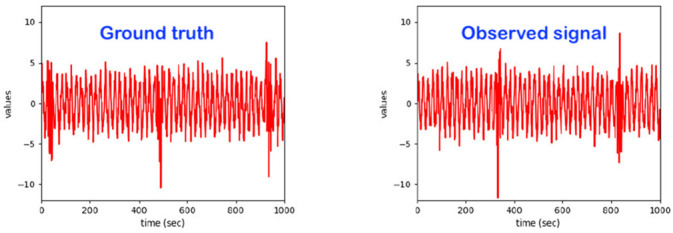	Here, one or more appear in the observed data.
Square pattern (AN-7)	Time interval observation	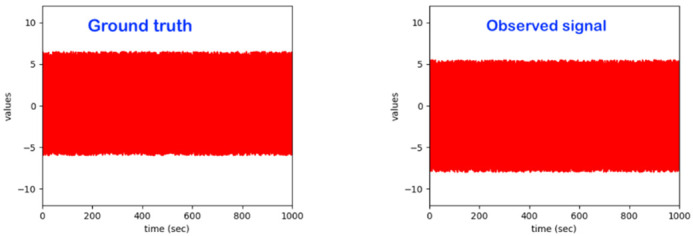	Here, the time response oscillates within a limited range, such as a square signal.
Trend pattern (AN-8)	Time interval observation	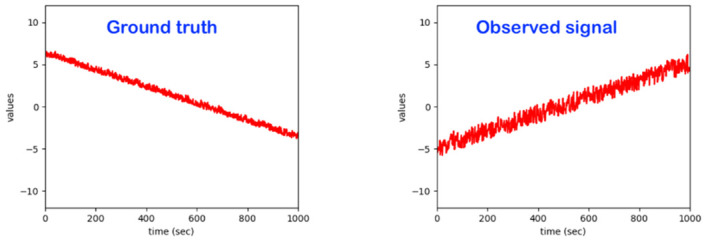	Here, the observed data/signal has an obvious non-stationary and monotonous trend.
Drift data pattern (AN-9)	Time interval observation	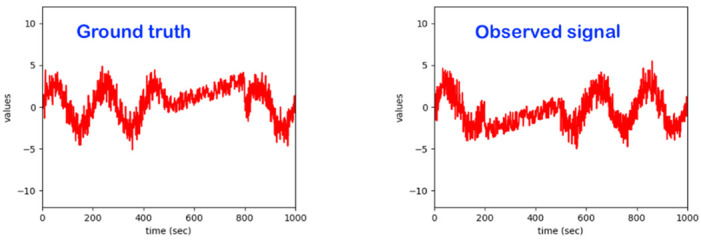	Here, the observed data signal is non-stationary with a random drift.

**Table 2 sensors-22-06279-t002:** Illustration of the time case (i.e., relative timing situation between detection and occurrence of the element to be assessed) perspective; see Figure 4.

Time of the Detection Process	Location (in the Time Dimension) of the Element to Be Assessed (See OCP 1 to OCP 4)	Time Case Labelling	Remarks and Eventual Comments
NOW (see Figure 4)	Sub-region 1/N	Online detection (Time case 2/a)	Here, the processing speed of the detection algorithm is critical. This depends on the computing infrastructure, namely embedded systems or processing in the cloud.
NOW (see Figure 4)	Sub-region 2/N	Near to online detection (Time case 2/b)	
NOW (see Figure 4)	Sub-region 3/N	Late online detection (Time case 2/c)	
NOW (see Figure 4)	Sub-region 1/P	Closest posterior detection (Time case 1/a)	This may be a new re-assessment of those past elements. This may be necessary in view of the time-varying system dynamics of the human under observation. It is theoretically possible that events that were perceived normal become later perceived abnormal after the system dynamics have evolved, and vice-versa.
NOW (see Figure 4)	Sub-region 2/P	Close posterior detection (Time case 1/b)	
NOW (see Figure 4)	Sub-region 3/P	Late posterior detection (Time case 1/c)	
NOW (see Figure 4)	Sub-region 1/F	Nearest anterior detection (Time case 3/a)	Here, one is looking into the future. In case one can see anomalies in that future, one has then a case of early warning: from a “close” early warning up to a “far” early warning situation. The elements to be assessed are coming from forecasting of the future system behavior at all levels in Figure 3.
NOW (see Figure 4)	Sub-region 2/F	Close anterior detection (Time case 3/b)	
NOW (see Figure 4)	Sub-region 3/F	Far anterior detection (Time case 3/c)	

**Table 3 sensors-22-06279-t003:** Temporal view of the novelty patterns—some illustrative examples (non-exhaustive).

Novelty Pattern Name and ID	Related Illustrative Graphical Illustration	Remarks
NOV 1: Sudden novelty	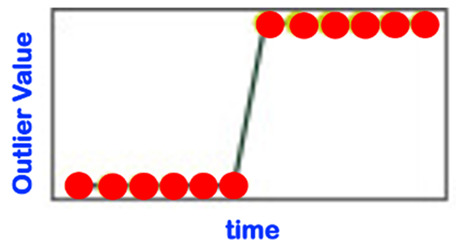	A sudden drift is characterized by an abrupt change in the underlying process (the one guiding the occurrence of anomalies over the time window observed).
NOV 2: Gradual drift	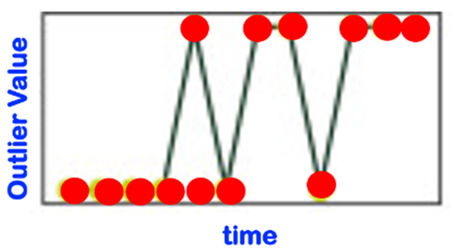	A gradual drift happens over time, and observations from one or more processes may be observed with changing frequency.
NOV 3: Incremental drift	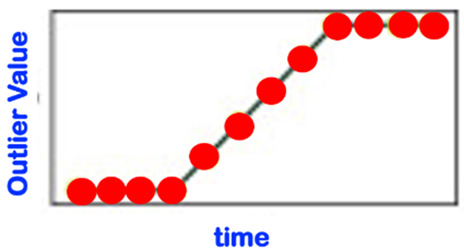	The incremental drift also happens over time but also involves one or more processes.
NOV 4: Reoccurring behavior/concept	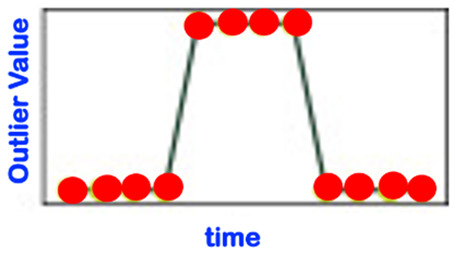	The reoccurring concept/behavior describes processes that cease to exist at one point and time but reappear later in time.

**Table 4 sensors-22-06279-t004:** Selected advanced characteristics of a robust and real-world mature abnormality detection system in the monitoring of humans in bed.

Selected Advanced Characteristics and Their Naming	Importance of a Real-World Mature Monitoring System	Remarks
**REQ 1**: Self-learning and continual learning capability, for either single individuals or groups of individuals	Very high	Continual learning can be understood as a concept of learning a model for a large number of “tasks” sequentially without forgetting knowledge obtained from the preceding tasks, whereby the data in/of the old tasks are not available anymore while training new ones. The learning relates to either the behavior of an individual person currently under monitoring or to the behavior of an individual person within the contextual background of the behaviors of several other persons (a group of persons) who have been priorly monitored.
**REQ 2**: Identification of and adaptivity to novelty/evolution	Very high	After the novelty has been identified, the model adapts to the confirmed behavior change. This means essentially adjusting to change over time.
**REQ 3**: Comprehensive uncertainty model/assessment for all subtasks (A–E) described in Section 3.4.	Very high	Anomaly detection in the case of monitoring a human in bed requires a high level of trust in its results. A key to this trust is the ability to assess the uncertainty of the computed results appropriately.
**REQ 4**: Prediction capability of the system status at levels 1 to 4 of Figure 3 for one or more future time sub-regions	Very high	See Subtask E described in Section 3.4.
**REQ 5**: Reconfigurability w.r.t. USPs (user-specific perspectives)	Very high	Since the USP are practically of high relevance, related reconfigurability of the intelligent system is needed.
**REQ 6**: Reconfigurability w.r.t. OCPs (observation context perspectives)	Very high	Since the OCP are practically of high relevance, related reconfigurability of the intelligent system is needed.
**REQ 7**: Explainability of the identification of entities at levels 1–4 of Figure 3	Very high/MUST	To avoid the lack of interpretability, this characteristic is needed.
**REQ 8**: Explainability of the anomaly detection (considering USPs and OCPs)	Very high	To avoid the lack of interpretability, this characteristic is needed.
**REQ 9**: Explainability of the evolution detection	Very high	To avoid the lack of interpretability, this characteristic is needed.
**REQ 10**: Explainability of the uncertainty grade or confidence level for all Sub-tasks (A–E) described in Section 3	Very high/MUST	To avoid the lack of interpretability, this characteristic is needed.
**REQ 11**: The possibility of a performance tuning/improvement through either partial human assistance (via some form of feedback) or evolutive/reinforced learning or involving artificially generated data out of some reliable “generative adversarial” process or a combination of some or of all of the above.	Very high/MUST	The trained personnel that operates the advanced intelligent monitoring system can, through an appropriate human–machine interface, confirm or inform some of the predictions/detections. Alternatively, self-learning triggered reinforcement learning can also be used.
**REQ 12**: The reconfigurability w.r.t. tolerance level/grade/margin related to the abnormality assessment (or, in other words, related to the anomaly detectability). The operators of the intelligent system for anomaly detection in the monitoring of “human in bed” is able to modify, even dynamically, the tolerance level/margin of the anomaly detection. This characteristic complements the so-called USPs described in Section 3.4; see also REQ 5.	Very high/MUST	For most technical systems, the acceptable tolerance level w.r.t. key system parameters is very important and very sensitive from a practical point of view. Indeed, in real-world applications and practice, in general, a low tolerance margin may result in a significantly much more expensive system. A bigger tolerance is thus resulting in a more interesting cost/benefit ratio. It is evident that the abnormality detection endeavor can therefore not ignore the tolerance level dimension. This is especially very sensitive in view of two critical facts related to the human monitoring scenario: (a) the sensor data obtained from the Level 0 of the architecture shown in Figure 3 are surely never perfect and thus full of uncertainties originating from various pure sensor systems related imperfections and disturbances; (b) the time-variant and stochastic physical activity related behavior pattern of the system “human in bed”; (c) the variance w.r.t. behavior pattern amongst different samples of the system under observation (i.e., the “human in bed”). This characteristic surely also impacts the interpretation of the anomaly score. In practice, the use-case engineer should know or at least be able to fix how much tolerance margin is really needed. Indeed, it makes no sense to fix/set a very small tolerance margin (which is very expensive to realize), although the given use-case can well be satisfied by a much bigger tolerance margin. In the various and extensive experimentation in the frame of a comprehensive verification process, it is worth closely studying the sensitivity of the tolerance margin w.r.t. the robustness of the intelligent anomaly detection system.
**REQ 13**: Tolerance to non-ideal training data	VERY HIGH/MUST	This is one practical requirement expressed by P.SYS. Essentially, data imperfection have several faces: (a) sensors related imperfections such as low update rate, low accuracy, noise and/or bias in the data, signal-related faults, etc.; (b) sensor’s drifts and/or other nonlinear, eventually time-varying disturbing phenomena; (c) data size related imperfections (this may be related to the effective (short) duration of the data recordings (i.e., observations and/or to the number of human samples involved, etc.; (d) involving only one of maximum two sensor modes or types (e.g., solely piezo-electric sensor and vibration sensors).
**REQ 14**: The complete intelligent system is **capable of running fully on COTS embedded platforms** (i.e., the intelligent system is fully Embedded AI), which are essentially low-computing power; this to ensure both low cost and data security while satisfying use-case specific real-time processing deadlines.	HIGH or LOW/MUST (A) VERY HIGH for a significant part of the target application scenarios. (B) “LOW” for application scenarios that wish or must involve IoT technologies	This requirement ensures that the intelligent system is low cost, has a relatively low power consumption, is application-scenario-dependently real-time capable and can operate almost self-sufficiently without involving remote computing infrastructure(s) and/or data. However, in the age of Cloud Computing and IoT, another part of the application scenarios may wish to enjoy the benefits of these recent, very advanced infrastructures. For these parts of the applications universe, the importance and criticality of this requirement (REQ 14) are rather low. Under the hypothesis (which is, however, not yet fulfilled at present) that “data security” is well and strong-reliably ensured in an “IoT and cloud computing” based networked universe, the core benefits of the “fully embedded system” version (which are essentially: low cost, low power consumption (by the end devices), real-time processing) become also fully satisfied by an Intelligent System architecture involving IoT and Cloud computing.
**REQ 15**: Short learning duration and/or fast detection of/and adaptation to behavior changes	VERY HIGH/MUST	This requirement also integrates, additionally, the fast detection of behavior change. It factually complements REQ 2. An issue here is, however, to clearly specify what “short” or “fast” means in the context of this requirement. This may be use-case dependent, as the use-case determines the basic time constant of the system. For example, some hours, some days, some weeks, etc. The effective “learning duration” length may/shall impact or strongly correlate with the performance metrics “Anomaly Detectability (MET-4)” that is described further below in Table 5.

**Table 5 sensors-22-06279-t005:** A selected collection, just for illustrative) of performance evaluation metrics (METs), which can be relevant for a comprehensive verification and validation of an anomaly detection-and-prediction system developed for the monitoring of a human in bed.

Metric Name and ID	Metric Description	Remarks
MET 1: Accuracy	It is simply defined as the mean squared error (MSE) between the model’s predictions and the target values.	Although this metric is named “accuracy”, it is actually a measure of error, and a low value is desired.
MET 2: Self-sensitivity	For self-associative empirical models, a robust model does/shall produce small changes in all of its outputs for (in the face of) small errors in the (model) inputs.	The self-sensitivity is a measure of an empirical model’s ability to make correct anomaly predictions when the respective anomaly-related score value is incorrect due to some sort of uncertainty (or fault).
MET 3: Cross-sensitivity	Cross-sensitivity measures the effect a faulty (model) input has on the other (model) predictions.	
MET 4: Anomaly detectability	This metric help to determine the smallest drift (in the relevant input data values of the detection system) that can be identified. Therefore, this anomaly detection performance metric is used to determine the smallest process parameter change that can be detected.	
MET 5: Precision	The precision answers the question: “What proportion of identified anomalies are true anomalies?”	This is a classical metric
MET 6: Recall	The recall is used to answer the question: “What proportion of true anomalies was identified?”	This is a classical metric
MET 7: F1 Score	The F1 score identifies the overall performance of the anomaly detection model by combining both recall and precision, using the harmonic mean.	This is a classical metric

**Table 6 sensors-22-06279-t006:** Discussion of comprehensive main seven reasons [48] that make graph-based approaches to anomaly detection vital and necessary.

Reason Supporting the Use of Graph Networks	Explanation	How Far Is It Relevant for Our Target Context of Monitoring a Human in Bed
Strong inter-dependence between entities and data	Data objects are often related to each other and exhibit dependencies. In fact, most relational data can be thought of as inter-dependent, which necessitates accounting for related objects in finding anomalies.	Highly relevant
Powerful representation ability	Graphs naturally represent the inter-dependencies by the introduction of links (or edges) between the related objects. The multiple paths lying between these related objects effectively capture their long-range correlations. Moreover, a graph representation facilitates the representation of rich datasets enabling the incorporation of node and edge attributes/types.	Highly relevant
The relational nature of problem domains	The nature of anomalies could exhibit itself as relational. An illustration example can be given from the performance monitoring domain, where the failure of a machine could cause the malfunction of the machines dependent on it. Similarly, the failure of a machine could be a good indicator of the possible other failures of machines in close spatial proximity to it (e.g., due to an excessive increase in humidity in that particular region of a warehouse).	Highly relevant
Graphs are a robust machinery	One could argue that graphs serve as more adversarial robust tools. For example, in fraud detection systems, behavioral clues such as log-in times and locations (e.g., IP addresses) can be easily altered or faked by advanced fraudsters. On the other hand, it may be reasonable to argue that the fraudsters could not have a global view of the entire network (e.g., money transfer, telecommunication, email, review network) that they are operating. As such, it would be harder for a fraudster to fit into this network as good as possible without knowing its entire characteristic structure and dynamic operations.	Highly relevant
Dynamic Graphs offer unique capabilities for anomaly detection	The anomaly detection in dynamic graphs can be based on the following situations: feature-based events, decomposition-based events, community or clustering-based events and window-based events.	Highly relevant(for example, see some of the OCPs)
Strong graph-based anomaly description capability	This is underscored by the following capabilities that are well documented in the relevant literature: (a) interpretation-friendly graph anomaly detection; (b) interactive graph querying and sense-making.	Highly relevant (see REQ 7 to REQ 10 in Table 4)
Several proven application examples of graph-based anomaly detection in highly complex real-world applications	Following applications examples can be found in the relevant literature: anomalies in telecom networks; anomalies in opinion networks; anomalies in auction networks; anomalies in the web network; anomalies in account networks; anomalies in social networks; anomalies in security networks; anomalies in computer networks; anomalies in financial networks	Highly relevant

**Table 7 sensors-22-06279-t007:** A comprehensive overview of the four major approaches or paradigms for anomaly detection and/or forecasting, which are also an application for the target context of “monitoring the physical activity of a human in bed”.

Paradigm- Identifi-Cation	Core Quintessence of the Paradigm	Selected Representative Related Works Related to Anomaly Detection and/or Prediction
MAJA 1: Statistical or “stochastic processes” based methods	A good representative of these methods are the so-called hidden Markov models (HMM). HMMs are statistical models to capture hidden information from observable sequential symbols/values. In an HMM, the system being modeled is assumed to be a Markov process with unknown parameters, and the challenge thereby is to determine the hidden parameters from the observable parameters. HMMs are sequence models. Thus, given a sequence of inputs, an HMM computes a sequence of outputs of the same length. An HMM model is a graph where nodes are probability distributions over labels, and edges give the probability of transitioning from one node to the other. Together, these can be used to compute the probability of a label sequence given the input sequence. By using HMM, it is possible to predict future states based on the current observations as well as the sequence of states from an observed sequence. For a process under observation over time, the possible states, which are hidden parameters, are generally “normal”, “abnormal” and “critical”.	Forkan et al. (2014) [26]; Girdhar, Mansi, et al. (2021) [53] Note: After appropriate tuning, the concept presented in these selected references can be used to model the scene described in Figure 3, at least for some of the defined OCPs and USPs. Each OCP is displayed by a different HMM model. Multiple OCPs can be considered simultaneously, resulting in much more complex HMM architectures.
MAJA 2: Deep-learning and neural networks (DL) based methods	Deep-learning (DL) concepts use complex neural networks for modeling time series and are thereby capable of detecting and/or predicting anomalies. DL models are very good at modeling the “temporal context” of a dynamically evolving system. The family of DL concepts of relevance for anomaly detection/prediction is well represented by five core models, which are: RNN (recurrent neural network), CNN (convolutional neural network), HYBRID (that is, a merging of the two previous ones), ATTENTION (refers to the so-called attention-based models), HTM (hierarchical temporal memory) and HTM. Each of these models displays specific interesting features w.r.t. the capability to capture the temporal context: (a) CNN: recognizes pattern sequences and predicts expected values, determines anomalies by identifying the differences between the predicted and actual signals, learns long-term dependencies by determining the number of previous states to keep or forget at every time step, extract multi-scale features while modeling long-term dependencies. CNN: instead of explicitly capturing the temporal context, it learns patterns in segmented time series; in order to comprehend behaviors appearing over a long period, a temporal convolution is used; three properties of the temporal convolutions: (i) they are causal, meaning that they ensure no information leakage from the future to the past; (ii) they can take a sequence of any length, just as with an RNN; (iii) they can look quite far into the past to forecast futures. HYBRID: when monitoring time-series data with a sliding window, the detectable anomaly pattern varies according to the window size; it considers the spatial information and temporal dependencies simultaneously; it solves the solve the spatiotemporal sequence-forecasting problem; eventually, a temporal “attention mechanism” adjusts the contribution of the previous feature maps to update the current one. The so-called “attention mechanism” was introduced to improve the performance (this it is a tuning) of the encoder-decoder model for machine translation. The idea behind the attention mechanism was to permit the decoder to utilize the most relevant parts of the input sequence in a flexible manner by a weighted combination of all of the encoded input vectors, whereby the most relevant vectors are attributed the highest weights. ATTENTION: by paying attention to the input weights that contribute more to the output, the so-called attention-based models can capture a very long-range dependence with relative importance to each data point; its remarkable achievements in NLP (natural language processing) motivated an application also in the time-series anomaly detection domain. HTM: it can reliably capture and predict sequence patterns and thus is beneficial to anomaly detection in time-series data; it is considered to be one of the most promising next-generation deep learning approaches; it is especially unique in that it does continuously learn temporal patterns from streaming data without backpropagation.	A sufficiently exhaustive inventory of schemes within this family of (DL-based) methods is provided in the survey by Kukjin Choi et al. (2021) [54]. **Consider RNN**, two examples of models: (a) LSTM family (e.g., LSTM-VAE, SPREAD, MAD-GAN), (b) GRU-gated recurrent unit) family (e.g., THOC, GGM-VAE, S-RNN). **Consider CNN**, two examples of models: (a) traditional CNN family (e.g., MU-Net, BeatGAN), (b) Temporal Convolutional Networks (TCU) family (e.g., HS-TCN, TCN-GMM). **Consider HYBRID**, one example of model: the Convolutional LSTM (ConvLSTM) family (e.g., MSCRED, RSM-GAN). **Consider ATTENTION**, one example of models: the self-attention or transformer family (e.g., MTSM, GAT). **Consider HTM**: one example of a model: RADM, a concept that integrates HTM and a Bayesian network [55].
MAJA 3: Graph-network-based methods, combined with either traditional ML (machine learning) methods or MAJA 1	Data objects representing a scene like the one in Figure 3 cannot always be treated as points lying in a multi-dimensional space independently. In contrast, they may exhibit inter-dependencies that should be accounted for during the anomaly detection process. Indeed, graphs provide powerful machinery for effectively capturing these long-range correlations amongst inter-dependent data objects. Anomaly detection methods for static graph data concern both unlabeled (plain) and labeled (attributed) graphs. Moreover, change or event detection approaches for time-varying or dynamic graph data, based for, e.g., on edit distances and connectivity structure, are available. Of particular interest is the so-called “anomaly attribution”, which consists of revealing the root cause of the detected anomalies and presenting anomalies in a user-friendly form. This provides tools that could/do enable/facilitate the post-analysis of detected anomalies for the crucial task of sense-making. Indeed, qualitative analysis techniques for the sense-making of spotted anomalies are very important and needed.	A sufficiently exhaustive inventory of schemes within this family of (graph-based) methods is provided in the survey by Leman Akoglu et al. (2015) [48]
MAJA 4: Graph neural network (Graph-DL) based methods	A very significant change in the last years is that graph anomaly detection (see MAJA 3) has evolved from relying heavily on the domain knowledge of human experts towards rather machine learning techniques that eliminate human intervention and, more recently, various deep learning technologies. These deep learning techniques are not only capable of identifying potential anomalies in graphs far more accurately than ever before, but they can also do so in real-time. Consequently, MAJA 4 can be viewed as a synergetic merging of MAJA 3 and MAJA 2. Moreover, certain bricks of MAJA 1 and of fuzzy logic may be easily casually integrated (e.g., Bayesian networks, certain HMM architectures, etc.). Conventional techniques (see MAJA 1 and MAJA 2) typically represent real-world objects as feature vectors and then detect outlying data points in the vector space. Although these techniques (see MAJA 1 and MAJA 2) showed power in locating deviating data points under tabulated data format, they inherently discard the complex relationships between objects. This discarding of relationships is a significant limitation especially, amongst others, from the perspective of “explainable anomaly detection and prediction” on hand and of the capability to model/map appropriately the complex scene of “monitoring a human in bed”, as was described in Figure 3 (while additionally considering the above so-called OCPs and USPs) on the other hand. Moreover, in real-world scenarios, many objects have rich relationships with each other, which can provide valuable complementary or dedicated special information for anomaly detection. Or the observation perspective from which the anomaly is detected can change even dynamically over time (see the above so-called USPs and OCPs). It is well known that non-deep-learning-based techniques (see MAJ-1 and MAJ-3) generally/essentially lack the capability to capture the non-linear properties of real objects [56]. Consequently, the representations of objects learned by them are not expressive enough to fully support a robust anomaly detection capability. In order to tackle these limitations, more recent studies seek the potential of adopting deep learning techniques to identify anomalous graph objects. By extracting expressive representations such that graph anomalies and normal objects can be easily separated, or the deviating patterns of anomalies can be learned directly through deep learning techniques, graph anomaly detection with deep learning is starting to take the lead in the forefront of anomaly detection. As a frontier technology, graph anomaly detection through integrated deep learning, hence, is expected to generate more fruitful results w.r.t. accuracy and robustness in detecting anomalies.	A sufficiently exhaustive inventory of schemes within this family of (graph DL-based) methods is provided in the survey by Xiaoxiao Ma et al. (2021) [47].

**Table 8 sensors-22-06279-t008:** A deep and comparative analysis of how far the four major approaches are capable or not of modeling and address the tough specifications expressed in the requirements dossier of the target context of “monitoring a human in bed”.

Requirement ID and Description	Capability of MAJA 1 to Fulfil the Requirement [57,58,59,60,61,62,63]	Capability of MAJA 2 to Fulfil the Requirement [64,65,66,67,68,69]	Capability of MAJA 3 to Fulfil the Requirement [48,70,71]	Capability of MAJA 4 to Fulfil the Requirement [69,72,73,74,75]	Remarks
REQ 1: Self-learning and continual learning capability, for either single individuals or groups of individuals	Possible but relatively/ eventually limited	Possible	Possible but relatively limited	Possible	All approaches can handle this requirement
REQ 2: Identification of and adaptivity to novelty/evolution	Possible but relatively/ eventually limited [26,37]	Possible [41,76]	Possible but relatively/ eventually limited [77]	Possible (a) [78] (b) [79]	All approaches can handle these requirements, whereby DL involving ones are better
REQ 3: Comprehensive uncertainty model/assessment for all subtasks (A–E) described in Section 3.4.	Not possible	Not possible	Possible but relatively/ eventually limited	Possible [80,81,82,83,84]	Only graph-based approaches can handle this requirement
REQ 4: Prediction capability of the system status at levels 1 to 4 of Figure 3 for one or more future time sub-regions	Possible but relatively/ eventually limited	Possible	Possible but relatively/ eventually limited	Possible [85,86,87,88,89]	All approaches can handle these requirements, whereby DL involving ones are better
REQ 5: Reconfigurability w.r.t. USPs (user-specific perspectives)	Not possible	Not possible	Eventually possible but very limited	Possible [90,91,92,93]	Only graph-based approaches can handle this requirement
REQ 6: Reconfigurability w.r.t. OCPs (observation context perspectives)	Not possible	Not possible	Eventually possible but very limited	Possible [90,91,92,93]	Only graph-based approaches can handle this requirement
REQ 7: Explainability [94] of the identification of entities at levels 1–4 of Figure 3	Not possible	Eventually possible but very limited	Eventually possible but very limited	Possible [95,96,97,98,99,100,101,102,103]	Only graph and/or DL-based approaches can handle this requirement
REQ 8: Explainability of the anomaly detection (considering USPs and OCPs)	Not possible	Not possible	Eventually possible but very limited	Possible	Only graph and/or DL-based approaches can handle this requirement
REQ 9: Explainability of the evolution detection	Not possible	Eventually possible but very limited	Eventually possible but very limited	Possible	Only graph and/or DL-based approaches can handle this requirement
REQ 10: Explainability of the uncertainty grade or confidence level for all subtasks (A–E) described in Section 3	Not possible	Eventually possible but very limited	Eventually possible but very limited	Possible	Only graph and/or DL-based approaches can handle this requirement
REQ 11: The possibility of a performance tuning/improvement through either partial human assistance (via some form of feedback) or evolutive/reinforced learning or involving artificially generated data out of some reliable “generative adversarial” process or a combination of some or all of the above.	Not possible	Eventually possible but very limited	Eventually possible but very limited	Possible [104,105,106,107]	Only graph and/or DL-based approaches can handle this requirement
REQ 12: The reconfigurability w.r.t. the tolerance level/grade/ margin related to the abnormality assessment (or, in other words, related to the anomaly detectability). The operators of the intelligent system for anomaly detection in the monitoring of “human in bed” shall able to modify, even dynamically, the tolerance level/margin of the anomaly detection.	Not possible	Eventually possible but very limited	Eventually possible but very limited	Possible	Only graph and/or DL-based approaches can handle this requirement
REQ 13: Tolerance to non-ideal training data	Possible (however, some adaptations may be necessary)	Possible (however, some adaptations (e.g., in the form of pre-processing layers or dataset augmentations) may be necessary)	Eventually possible but after adaptations	Surely possible	Almost all approaches can handle this requirement, although adaptations, which may be very substantial, may be necessary
REQ 14: The complete intelligent system shall be capable of running fully on COTS embedded platforms (i.e., the intelligent system shall be fully Embedded AI), which are essentially low-computing power; this is to ensure both low cost and data security while satisfying use-case specific real-time processing deadlines.	Possible (however, some adaptations may be necessary)	Possible (however, some significant architecture and pipeline adaptations may be necessary)	Eventually possible (however, some significant architecture and pipeline adaptations may be necessary)	Eventually possible (however, some significant architecture and pipeline adaptations may be necessary)	For this requirement, the approaches involving DL are not superior to the intelligent system version that is “fully embedded”. For this system version, MAJA 1 is potentially superior. However, the situation significantly changes for the case of an intelligent system that can/does involve IoT and related infrastructure such as Cloud Computing
REQ 15: Short learning duration and/or fast detection of/and adaptation to behavior changes	Possible (however, some adaptations may be necessary)	Possible (however, some significant architecture and pipeline adaptations may be necessary)	Possible (however, some significant architecture and pipeline adaptations may be necessary)	Possible (however, some significant architecture and pipeline adaptations may be necessary). However, more flexible in the presence of reconfigurability needs w.r.t. REQ 11 and REQ 12	The performance metric MET 4 (see Table 5) can influence (subject-dependently) this requirement. For non-reconfigurable setups, MAJA 1 appears potentially superior. However, in the face of reconfigurability needs, MAJA 4 becomes evidently the superior one.

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
