# Peer review of "A Comprehensive “Real-World Constraints”-Aware Requirements Engineering Related Assessment and a Critical State-of-the-Art Review of the Monitoring of Humans in Bed"

_sensors, 2022, doi:10.3390/s22166279_

Round 1
Reviewer 1 Report
It is a dense job, in which they have elaborated a large amount of data, so I see that it is a serious job from the point of view of programming and obtaining information.
The questions that I have asked myself throughout the reading are very basic, since the introduction has been a bit confusing for me. It begins by talking about anomalies, but the big question is what is normal? With respect to what values are the values obtained with the sensors on the legs of the beds going to be contrasted? Or is it rather a study over time to then work on that information? what will be measured? what do the sensors measure? that is to say, I missed a context that helps to understand the reason for so much work. speaking of anomalies in general is too broad, which leads to confusion.
In line 75 w.r.t appears for the first time, it is unknown what these acronyms refer to.
I understand that the keywords should be shorter.
Author Response
See PDF file attached

Reviewer 2 Report
This paper presents a comprehensive formulation of requirements for the monitoring system of humans in bed. Moreover, a review of activity monitoring of a human in bed was carried out.
1. The literature on a "human in bed" monitoring system is almost null. Therefore, a review of Abnormality-Sensitive, Subject-Adaptive, Evolutive, and Early-Warning-Capable Activity Monitoring of humans in bed was carried out. This review was not accomplished since the paper's main topic is "monitoring of humans in bed". It might be interesting to split the paper into two particular topics: 1) an original research paper in the field of "monitoring of humans in bed" and 2) a review of the monitoring system of human activity.
2. The objectives of the paper were not defined in the introduction. Conclusions are vast, and this research study lacks objectivity.
3. It is not evident to identify the original contribution. Maybe the system requirements of Tab. 4 or the results of the proposed formulation. Moreover, the proposed comprehensive formulation for the monitoring system of humans in bed was not compared with approaches previously proposed in the literature to highlight the novel contributions. It is not evident to evaluate the novelty of the proposed research study in terms of the theoretical developments of the methodology or the novelty of the results.
Figures: the quality of the figures is poor. It is recommended to include vector graphs.
Author Response
See file attached

Round 2
Reviewer 2 Report
The authors have enhanced the overall quality of the manuscript. The original contribution of this "review manuscript" is relevant and it is highlighted in the abstract and introduction.
Author Response
Thank you very much for acknowledging the revision changes made, which also do clearly highlight the contribution of the paper.